

# Flow equations for disordered Floquet systems

Steven J. Thomson[1,2,3*], Duarte Magano[2,4,5] and Marco Schiró[2]

**1** CPHT, CNRS, Ecole Polytechnique, IP Paris, F-91128 Palaiseau, France
**2** JEIP, USR 3573 CNRS, Collège de France, PSL Research University,
11 Place Marcelin Berthelot, 75321 Paris Cedex 05, France
**3** Institut de Physique Théorique, Université Paris-Saclay, CNRS, CEA,
F-91191 Gif-sur-Yvette, France
**4** Instituto de Telecomunicações, Physics of Information and
Quantum Technologies Group, Portugal
**5** Instituto Superior Técnico, Universidade de Lisboa, Portugal

★ steven.thomson@polytechnique.edu

## Abstract

In this work, we present a new approach to disordered, periodically driven (Floquet) quantum many-body systems based on flow equations. Specifically, we introduce a continuous unitary flow of Floquet operators in an extended Hilbert space, whose fixed point is both diagonal and time-independent, allowing us to directly obtain the Floquet modes. We first apply this method to a periodically driven Anderson insulator, for which it is exact, and then extend it to driven many-body localized systems within a truncated flow equation ansatz. In particular we compute the emergent Floquet local integrals of motion that characterise a periodically driven many-body localized phase. We demonstrate that the method remains well-controlled in the weakly-interacting regime, and allows us to access larger system sizes than accessible by numerically exact methods, paving the way for studies of two-dimensional driven many-body systems.

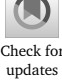
# 1 Introduction

Understanding the nonequilibrium dynamics of quantum many-body systems is a key challenge at the heart of modern research in condensed matter physics, motivated by a host of recent experimental developments in quantum simulation which have enabled unprecedented levels of control of strongly correlated quantum matter.

Experimental advances in ultracold atomic gases, for example, have made it possible to engineer almost perfectly isolated quantum systems and have allowed transport properties and nonequilibrium dynamics to be probed with a high degree of control and resolution [1]. One current frontier is the use of time-periodic drive, such as a laser or time-varying magnetic field, to engineer effective Hamiltonians, leading to new states of matter far from equilbrium [2–4]. This line of thought, known as Floquet engineering, has recently lead to a number of dramatic experimental breakthroughs with cold atoms in dynamically modulated optical lattices, including the realization of non-trivial topological phases [5,6], the control of magnetic corre-

lations in strongly interacting fermionic gases [7], and the experimental realization of strongly driven Fermi [8] and Bose Hubbard models [9,10], which play important roles in the quantum simulation of condensed matter systems.

Another exciting contemporary development is the experimental realization of novel phases of matter with no equilibrium counterpart, such as the Many-Body Localized (MBL) phase [11–14] seen in isolated disordered many-body quantum systems which fail to thermalize. Recent experimental advances in quantum simulators have allowed highly precise control over disordered many-body systems and led to evidence of MBL behavior in a number of platforms, ranging from one and two dimensional arrays of ultracold atoms [15–19] to ion traps with programmable random disorder [20,21] and dipolar systems made by nuclear spins [22,23].

The combination of disorder, interactions and periodic drive represent one of the current frontiers of the field. While ergodic quantum many-body systems are expected to reach thermal equilibrium of local observables at long times [24], which for driven systems lacking time translational invariance corresponds to infinite temperature [25, 26], MBL phases can avoid thermalisation even in presence of a periodic drive [27–29] and give rise to exotic nonequilibrium phases of matter such as quantum time crystals [30], which have been intensively theoretically explored in [31–40] and very recently experimentally observed [21, 41].

From a theoretical perspective, the study of periodically driven - or *Floquet* - quantum systems has a long history, in the context of phenomena ranging from dynamical localization [42] and quantum dissipation [43] to quantum chaos [44]. Due to the time-periodic nature of the Hamiltonian $H(t + T) = H(t)$ and the linearity of Schrodinger equation, the problem can be analyzed using the Floquet theorem, a basic result in the theory of linear ordinary differential equations and the time-analogue of Bloch's theorem in solid state physics. This formal analogy, which has been a major force in the recent understanding of driven quantum many-body problems [45], goes further with the introduction of an *effective Floquet Hamiltonian*, $H_F$, defined through the evolution operator over a period, $e^{-iH_F T} \equiv U(T, 0)$ where $T$ is the drive period. This static Hamiltonian can be explicitly derived, and is therefore of practical relevance, only in certain regimes such as at high drive frequency [46–49] where energy absorption is suppressed. In the more interesting regime of intermediate drive frequency one needs to go beyond the effective Floquet Hamiltonian [50, 51]. Numerical approaches based on exact diagonalization of the full evolution operator allow access to the complete information of the Floquet eigenstates and eigenmodes, but are usually limited to very small system sizes. In the context of disordered systems, powerful methods such as the strong-disorder renormalization group have been extended to the Floquet context [52–54].

In this work, we set out and demonstrate the use of the flow equation approach to study periodically driven and disordered many-body quantum systems. Flow equations have been used in recent years to study a wide variety of systems with time-independent Hamiltonians, including Kondo and impurity models [55,56], quenches in the Hubbard model [57], and more recently have gained a lot of attention in the context of quantum localization [58–65]. The method has also been successfully employed in the study of dissipative systems, either by decoupling the system from its environment (e.g. the spin-boson model studied in Refs. [66–68]) or more recently in a situation where the dynamics is generated by a Markovian Lindblad master equation [69] with a Lindbladian which can be diagonalised directly using this approach. In the context of Floquet systems previous works have used similar techniques to systematically derive effective Hamiltonians [70–72], and to study prethermal regimes [73]. Here instead we focus on the Floquet evolution operator and devise a flow equation approach to diagonalize it, therefore obtaining the quasienergies and Floquet eigenstates directly.

This paper is organized as follows. In Section 2 we first summarise the use of flow equations for static systems, discuss the generalisation of this approach to time-dependent systems and demonstrate why periodic drive is particularly amenable to study with flow equations. In

Section 3, we give a brief summary of Floquet theory and set out the mathematical formalism we will go on to use in Section 4 where we discuss a number of different flow equation approaches and show how our method differs from existing techniques. In Section 5 we present an example of the Floquet flow equation technique applied to a system of non-interacting fermions in a disordered potential subject to periodic drive. We then go on to show how the method may be extended to the case of fermions with weak interactions in Section 6, where we consider a driven many-body localized phase and show how our method gives insight as to the emergent local integrals of motion which characterise this phase, as well as an estimate for the breakdown of the localization as a function of drive frequency. Finally, we conclude with an outlook towards the future and discuss possible applications of our method beyond those which we present here.

## 2 The Flow Equation Method

Flow equation methods have a rich history as applied to time-independent models. They were originally introduced to condensed matter physicists by Wegner [74], independently in the context of high-energy physics by Glazek and Wilson [75,76] under the name of 'similarity transforms', and to mathematicians under the names 'double bracket flow' and 'isospectral flow' by Refs. [77–79]. For a detailed introduction to the method, we refer the reader to Refs. [56,74], but here we will present a brief overview of the original formulation of the method before discussing how it can be extended to time-dependent Hamiltonians.

### 2.1 Static Hamiltonians

In the case of a time-independent Hamiltonian, the flow equation formalism can be best explained by analogy with a Schrieffer-Wolff transform [80,81]. A Schrieffer-Wolff transform is a unitary transform of the form:

$$\tilde{H} = e^S H e^{-S} = H + [S, H] + ..., \tag{1}$$

where, by choosing $[S, H] = -V$ such that $V$ contains the off-diagonal terms, the Hamiltonian can be diagonalised to leading order. By expanding the exponential to higher orders the expansion can be made more accurate, at the cost of having to evaluate high-order commutators. Rather than making a single 'large' unitary transform, however, one can instead imagine making an infinitesimal transform which can be made arbitrarily accurate:

$$H(l + dl) = e^{\eta(l)dl} H(l) e^{-\eta(l)dl} = H(l) + dl \, [\eta(l), H(l)], \tag{2}$$

where $l$ is a fictional 'flow time' which runs from $l = 0$ to $l = \infty$ and $\eta(l)$ is some (scale-dependent) anti-Hermitian generator for the transform, which we shall discuss in detail later. By making infinitely many of these unitary transforms, the diagonalization process can be expressed as a single continuous unitary transform obeying the 'equation of motion':

$$\partial H(l) = [\eta(l), H(l)]. \tag{3}$$

By analogy with renormalization group, this equation is known as the 'flow equation' for the Hamiltonian. By integrating the flow equation, which is typically done numerically, one can obtain a diagonal Hamiltonian[1]. There are many possible choices for the generator $\eta(l)$, as the only requirements are that it is anti-Hermitian and diagonalises the Hamiltonian in the

---

[1]In principle, one can also construct the unitary transform explicitly as a time-ordered exponential $\tilde{U} = \mathcal{T}_l \int_0^\infty \exp(\eta(l)dl)$, however this is typically difficult to evaluate and rarely of practical use.

limit $l \to \infty$. In this work we shall concentrate on the canonical choice (also known as the 'Wegner generator' [74]) where we choose

$$\eta(l) = [H_0(l), V(l)], \tag{4}$$

where $H_0$ contains the diagonal components of the Hamiltonian and $V = H - H_0$ contains the off-diagonal terms. Other choices are possible [58,64], however the canonical generator tends to be a robust choice that can be stably numerically integrated. For reference, in Appendix A we sketch the application of this method to a system of non-interacting fermions where it can be applied straightforwardly and exactly.

Flow equations have proven to be useful non-perturbative tools able to access unique parameter regimes and system sizes inaccessible to most other state-of-the-art modern numerical methods, such as two-dimensional many-body localized systems or systems with disordered long-range couplings [61,65], and are the subject of active ongoing development due to their flexibility and range of desirable properties. One key advantage of the flow equation method over many other techniques is that flow equations treat all energy scales equivalently, and are not restricted to perturbative situations or low-lying excitations. As compared with renormalization group calculations, it is important to note that flow equations retain all information contained in the Hamiltonian at all stages of the flow, as they are simply unitary transforms of the original problem: no decimation is required and no information is lost, as the transform can always be reversed back into the original basis.

## 2.2 Time-dependent Hamiltonians

A formal extension of the flow equation method to time-dependent Hamiltonian problems was introduced in Ref. [82] in the context of the time-dependent Kondo model. We briefly recall the basic idea here, since it will serve as starting point for our Floquet flow. The aim is to perform a time-dependent unitary transformation into a frame in which the Schrödinger equation $i\partial_t |\psi(t)\rangle = H(t)|\psi(t)\rangle$ is simplified. The dynamics in the rotated frame, $|\tilde{\psi}(t)\rangle = U(t)|\psi(t)\rangle$, reads $i\partial_t |\tilde{\psi}(t)\rangle = \tilde{H}(t)|\tilde{\psi}(t)\rangle$ with a transformed Hamiltonian of the following form:

$$\tilde{H}(t) = U(t)[H(t) - i\partial_t]U^\dagger(t). \tag{5}$$

In the spirit of time-independent flow equations, a series of infinitesimal time-dependent unitary transformations is introduced, parametrized by the scale $l$ and with generator $\eta(l, t)$. The time-dependent analog of Eq. 3 becomes [82]:

$$\partial_l H(l, t) = [\eta(l, t), H(l, t)] + i\partial_t \eta(l, t). \tag{6}$$

This differs from the time-independent flow by the addition of the time derivative term, which vanishes for a time-independent problem and reduces back to Eq. 3 . Similarly, the canonical generator (Eq. 4) can be modified to:

$$\eta(l, t) = [H_0(l, t), V(l, t)] - i\partial_t V(l, t), \tag{7}$$

where $H_0(l, t)$ represents the diagonal part of the Hamiltonian and $V(l, t) = H(l, t) - H_0(l, t)$ represents the off-diagonal part. Following Ref. [82], this form of generator can be shown to eliminate all (non-resonant) off-diagonal couplings in the $l \to \infty$ limit, giving a diagonal, yet still time-dependent, Hamiltonian. However, for a general time-dependent protocol, this flow can be extremely complicated to solve, requiring the solution of a set of coupled partial differential equations in both flow time $l$ and real time $t$. In this work, we focus on situations where the time dependence of the microscopic Hamiltonian is periodic, and we find that in

this case the canonical generator (Eq. 7) can be implemented efficiently using Floquet theory (which we briefly introduce in the following section). This provides a starting point to formulate a flow equation approach directly in Floquet space which, differently from Eq. 7, eliminates both the off-diagonal terms as well as the time-dependence. We will discuss the details of this approach, which is the main result of this manuscript, in Section 4.

# 3 A Brief Introduction to Floquet Theory

## 3.1 Floquet states and quasi-energies

Periodically driven systems, known as Floquet systems, are described by Hamiltonians which are periodic in time, satisfying $H(t + T) = H(t)$ where $T$ is the period of the drive. Following Floquet's theorem [83], which is analogous to Bloch's theorem in solid state systems, periodically driven systems admit a complete set of quasi-periodic solutions of the time-dependent Schrödinger equation, as 'Floquet eigenstates'

$$|\Psi_\alpha(t)\rangle = e^{-i\varepsilon_\alpha t/\hbar} |\psi_\alpha(t)\rangle , \tag{8}$$

where $|\psi_\alpha(t + T)\rangle = |\psi_\alpha(t)\rangle$ are states with the same periodicity as the drive, which satisfy

$$(H(t) - i\partial_t)|\psi_\alpha(t)\rangle = \varepsilon_\alpha |\psi_\alpha(t)\rangle , \tag{9}$$

while the phases $\varepsilon_\alpha \in \mathcal{R}$ are known as the Floquet *quasi-energies*. The quasi-energies do not depend on the microscopic time $t$, only the period of the drive $T$, and are associated to corresponding Floquet eigenstates which form a complete orthonormal basis. Note that because of the complex exponential in Eq. 8, the quasienergies are only uniquely defined up to a shift by an integer multiple of $\omega = 2\pi/T$. By analogy with crystal momentum in solid-state systems, one typically defines a 'Brillouin zone' such that all quasienergies are uniquely determined in a given interval of width $\omega$. Here, we use the convention that $\varepsilon_n \in [-\omega/2, \omega/2] \ \forall \ n$. In the following, we shall work in units where $\hbar \equiv 1$. There are essentially two general approaches to obtain Floquet eigenstates and eigenmodes for quantum many-body systems, which we briefly sketch below.

## 3.2 Floquet Evolution Operator

In numerical studies based on exact solutions of quantum dynamics the most efficient way to obtain Floquet eigenstates and eigenvectors is through the evolution operator over a drive period,

$$U(T, 0) = \exp\left(-i \int_0^T dt H(t)\right), \tag{10}$$

which satisfies several useful properties. For example, the evolution at integer multiples of the drive satisfies $U(nT, 0) = (U(T, 0))^n$, which implies that as long as one observes the system only stroboscopically the quantum evolution corresponds to applying $n$ times the same evolution operator $U(T, 0)$, as for closed undriven systems. This leads to the introduction of an effective Floquet Hamiltonian associated to this fundamental evolution $U(T, 0) \equiv \exp(-iTH_F)$. Furthermore, one can show that the Floquet modes are the eigenstates of the evolution operator during a period $T$ with eigenvalues $e^{-i\varepsilon_\alpha T/\hbar}$ which suggests the decomposition

$$U(t + T, t) = \sum_\alpha e^{-i\varepsilon_\alpha T/\hbar} |\psi_\alpha(t)\rangle \langle\psi_\alpha(t)| . \tag{11}$$

The above results suggest several practical numerical approaches to solve the Floquet problem. One possibility involves computing the evolution operator from time $t = 0$ to time $T$, solving the exact dynamics, then diagonalizing it to find the Floquet modes at time $t = 0$ and the quasi-energies. Then one can just propagate these states to get them in the full first cycle. Otherwise one can solve for the evolution operator in the full first cycle $U(t, 0)$ (where $t$ is within the first period) and then diagonalize it to obtain directly the Floquet modes over the first cycle. While these approaches are practically useful for exact numerics, here we choose a different path for our Floquet flow equation approach using the expansion in an extended Hilbert space.

### 3.3 Extended Floquet Hilbert space

Since the Floquet modes are periodic with period $T$ it is possible to expand them in terms of Fourier harmonics at integer multiples of their fundamental frequency $\omega$, and to associate each of these harmonics with an element the space of $T$-periodic, square-integrable functions $\mathcal{L}_T$. This results into an enlarged Floquet Hilbert space (often referred to as a Sambe space [84]) denoted $\mathcal{F} = \mathcal{H} \otimes \mathcal{L}_T$ which is given by the tensor product of the original Hilbert space of the microscopic system $\mathcal{H}$ and $\mathcal{L}_T$.

Explicitly, for a given time-periodic Floquet mode $|\psi_\alpha(t)\rangle$ in $\mathcal{H}$, we can write it in the extended Floquet Hilbert space $\mathcal{F}$ by:

$$|\psi_\alpha(t)\rangle = \sum_n |\psi_\alpha^n\rangle \, e^{in\omega t} = \sum_n |\psi_\alpha^n\rangle \otimes |n\rangle \,, \tag{12}$$

which motivates the introduction of $\hat{\sigma}_n |m\rangle = |n + m\rangle$ as a creation operator in 'frequency space' $\mathcal{L}_T$ associated to the $n^{\text{th}}$ harmonic, and the sum over harmonics runs over both positive and negative values. The index $n$ is often known as the 'photon number' by analogy with the quantum systems driven by coherent radiation [85], although here we shall refer to it as the 'harmonic index'. Plugging this expansion in Eq. (9) allows us to cast the problem of finding Floquet modes into a conventional eigenvalue problem in a higher dimensional space, a sort of synthetic extra dimension [86, 87]. To see this we notice that the time derivative $-i\partial_t$ acting in $\mathcal{H}$ can be rewritten in the extended Hilbert space $\mathcal{F}$ as:

$$\mathcal{D} = \mathbb{1} \otimes \omega \hat{n} \,, \tag{13}$$

where $\hat{n} = \sum_n n |n\rangle \langle n|$ is the number operator in frequency space. This can be verified by showing that $-i\partial_t \exp(in\omega t) = \omega n \exp(in\omega t)$, and therefore $-i\partial_t \to \omega \hat{n}$ in $\mathcal{L}_T$. The main object of interest in this extended Hilbert space is known as the Floquet quasienergy operator [88], given by:

$$K = H(t) - i\partial_t = \sum_n H^{(n)} \otimes \hat{\sigma}_n + \mathbb{1} \otimes \omega \hat{n} \,, \tag{14}$$

whose eigenstates and eigenvalues give the Floquet modes. As noted in Ref. [88], although the quasienergy operator $K$ lives in a dramatically larger Hilbert space than the initial formulation of the problem, there is a considerable simplification involved in that we can now use standard time-independent methods to diagonalize $K$. Moreover, we never need to constuct $K$ explicitly, as we shall go on to show: this is important as the explicit form of $K$ contains a lot of redundant information due to the tensor product structure. In Section 4 we will discuss several different strategies to diagonalize $K$, before showing a concrete example in Section 5 for a non-interacting system, and demonstrate the extension to weak interactions in Section 6.

# 4 Flow Equations for Floquet Systems

Based on the discussion of previous sections we can now present the main aim of this manuscript, which is to devise a flow equation approach to diagonalize the Floquet operator $K$ in the extended Floquet Hilbert space. This is achieved by applying a continuous unitary transform parameterized by a running scale $l$ and with generator $\eta(l)$ such that the flow of $K$ towards a diagonal form is given by

$$\mathrm{d}K/\mathrm{d}l = [\eta(l), K(l)], \tag{15}$$

for a suitable choice of $\eta$. In the static case, the choice of unitary transform is not unique, and one has a lot of freedom to choose a generator with properties which suit the problem. For example, it is possible to choose a generator which preserves the sparsity of the Hamiltonian [58], although this turns out to be numerically difficult to integrate [64]. In the case of driven systems, however, there is an even richer variety of possible choices. Depending on the requirements, one may choose a generator which diagonalises $K$ in $\mathcal{H}$ (i.e. a diagonal, but still time-dependent Hamiltonian), in $\mathcal{L}_T$ (time-independent but not diagonal in real space) or in the full Hilbert space $\mathcal{F}$. Here we review several choices from the existing literature which accomplish the first two goals, and propose a new generator which cuts to the chase by acting directly in $\mathcal{F}$ to produce a diagonal, time-independent Floquet operator from which we can directly obtain the quasi-energies and Floquet eigenstates.

In each of the following cases, we shall split the Floquet quasienergy operator $K$ into a diagonal piece $K_0$ and an off-diagonal piece $K_{\mathrm{off}}$, and choose the generator to be $\eta = [K_0, K_{\mathrm{off}}]$. We have the complete freedom to choose the form of these terms to be whatever we wish: we shall see that by making different choices, we can obtain qualitatively very different results. From here on, we suppress explicit dependence on the running scale $l$ for clarity.

## 4.1 Type I: Canonical Generator

The canonical generator (Eq. 7), corresponds to the time-dependent generator introduced in Ref. [82], which we may expand in Floquet harmonics. As such, it diagonalises the Hamiltonian in space, but leaves the final form time-dependent, i.e. diagonal in $\mathcal{H}$ but not in $\mathcal{L}_T$. The result of this flow is shown schematically in Fig. 1b. Concretely, in order to compute the unitary transform using this generator, we split the Floquet quasienergy operator $K$ into diagonal ($K_0$) and off-diagonal ($K_{\mathrm{off}}$) pieces:

$$K = \underbrace{\sum_n \sum_i H_{ii}^{(n)} |i\rangle \langle i| \otimes \hat{\sigma}_n + \mathbb{1} \otimes \omega \hat{n}}_{K_0} + \underbrace{\sum_n \sum_{i \neq j} H_{ij}^{(n)} |i\rangle \langle j| \otimes \hat{\sigma}_n}_{K_{\mathrm{off}}}, \tag{16}$$

where the canonical generator Eq. 7 is then given by $\eta = [K_0, K_{\mathrm{off}}]$. This amounts to the choice that the 'off-diagonal' elements to be removed by the unitary transform are the terms which are off-diagonal in real space, regardless of their time-dependence.

In order to obtain the quasi-energies, the time-dependence of the final result must be removed by a further change of basis into a rotating frame such that the problem becomes time-independent.

## 4.2 Type II: Verdeny-Mielke-Mintert

It is also possible to use flow equations to obtain a time-independent effective Floquet Hamiltonian which is not diagonal in real space. To do this, we may make a different decomposition

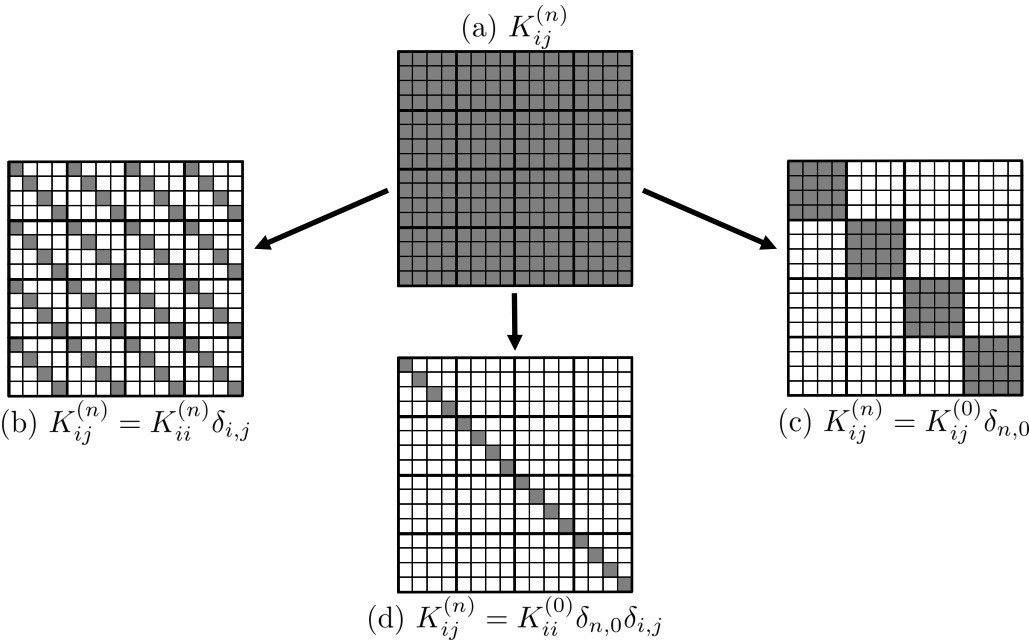

Figure 1: A schematic of different flow schemes for the Floquet quasienergy operator, here denoted as $K = \sum_{i,j,n} K_{ij}^{(n)} |i\rangle \langle j| \otimes \hat{\sigma}_n$. Panel (a) shows the full Floquet quasienergy operator $K$ - note that in all of the cases we consider, this is a sparse matrix, however the method can equally be applied to dense matrices. Panels (b), (c) and (d) show the form of the diagonal matrix after the application of Type I, II and III generators respectively. Panel (b) shows that Type I generators (Section 4.1) result in a Floquet matrix which is time-dependent but diagonal in space, while (c) demonstrates that Type II generators (Section 4.2) result in a block-diagonal form of $K$ which is time-independent but not diagonal in real space. Panel (d) shows the result after applying a Type III (Section 4.3) generator which we focus on in the rest of this manuscript, which is both diagonal in space *and* time-independent.

of the Floquet operator into diagonal and off-diagonal parts, following Ref. [70]:

$$K = H^{(0)} \otimes \mathbb{1} + \underbrace{\mathbb{1} \otimes \omega \hat{n}}_{K_0} + \underbrace{\sum_{n \neq 0} H^{(n)} \otimes \hat{\sigma}_n}_{K_{\text{off}}}, \tag{17}$$

which, in contrast with the previous case, amounts to choosing the 'off-diagonal' terms which will be eliminated by the flow to be all terms with harmonic index $n \neq 0$ regardless of their real-space structure. With this choice, the generator becomes:

$$\eta = [K_0, K_{\text{off}}] = [\mathcal{D}, K_{\text{off}}] = \sum_n \omega n \sum_{ij} H_{ij}^{(n)} |i\rangle \langle j| \otimes \hat{\sigma}_n. \tag{18}$$

Contrary to the canonical generator, this choice leads to a Hamiltonian which is time-independent without being diagonal in space, i.e. diagonal in $\mathcal{L}_T$ but not in $\mathcal{H}$. In figure Fig. 1c we show a pictorial representation of this type of flow - it leaves $K$ block-diagonal, i.e. diagonal in frequency (time-independent) but not diagonal in space. This is useful if one wishes to obtain the effective Floquet Hamiltonian $H_F$ for stroboscopic evolution, or study the form of the effective Hamiltonian in the context of Floquet engineering. This is similar in spirit to the method of Ref. [72] designed to construct effective Floquet Hamiltonians, although the

precise choice of generator is different. If one is interested in the quasi-energies and Floquet eigenstates then one must then employ a second transformation to diagonalize $H_F$.

### 4.3 Type III: Wegner-Floquet Flow

Motivated by the two previous choices, we here propose a modified generator combining aspects of both which we call the 'Wegner-Floquet' (WF) generator. Essentially, it is given by Wegner's canonical choice (Eq. 4) *applied to the Floquet quasienergy operator* (i.e. acting in $\mathcal{F}$) rather than to the bare Hamiltonian (i.e. acting only in $\mathcal{H}$). The end result is that the WF generator will completely diagonalize the problem in the full extended Hilbert space $\mathcal{F} = \mathcal{H} \otimes \mathcal{L}_T$. The crucial difference is that in this choice of generator, we choose the 'off-diagonal' terms to include both the off-diagonal terms in $\mathcal{H}$ as well as the $n \neq 0$ harmonics of all terms ('off-diagonal' in $\mathcal{L}_T$), fulfilling our goal of finding a generator which can directly obtain the quasi-energies. A schematic representation is shown in Fig. 1d, where the final form of $K$ is diagonal in both space and time.

We write the Floquet quasienergy operator as:

$$K = \underbrace{\sum_i H_{ii}^{(0)} |i\rangle \langle i| \otimes \mathbb{1} + \mathbb{1} \otimes \omega \hat{n}}_{K_0} + \underbrace{\sum_i \sum_{n \neq 0} H_{ii}^{(n)} |i\rangle \langle i| \otimes \hat{\sigma}_n + \sum_{i \neq j} \sum_n H_{ij}^{(n)} |i\rangle \langle j| \otimes \hat{\sigma}_n}_{K_{\text{off}}}, \quad (19)$$

where we explicitly separate the zero-frequency diagonal components ($K_0$, which becomes the Floquet quasi-energies in the limit $l \to \infty$) from the rest of the Floquet matrix, defining $K_{\text{off}}$ as all other components (comprising finite-frequency terms which are diagonal in real space, and all off-diagonal terms at all frequencies). As before, we choose the generator to be $\eta = [K_0, K_{\text{off}}]$, and the end result of our diagonalization process will be a matrix of the form

$$\tilde{K} = \tilde{H} \otimes \mathbb{1} + \mathbb{1} \otimes \omega \hat{n}, \quad (20)$$

where $\tilde{H} = \sum_i \tilde{h}_i^{(0)} n_i$ are the Floquet quasi-energies.

To obtain an explicit expression for the flow it is useful to start from Eq. (19) for the Floquet operator, write it in terms of matrix elements in the Hilbert and harmonic space and separate the diagonal and off-diagonal components according to the Wegner-Floquet generator, i.e.

$$K_0 = \sum_i H_{ii}^{(0)}(l) |i\rangle \langle i| \otimes \mathbb{1} + \omega \sum_i |i\rangle \langle i| \hat{n} \quad (21)$$

and

$$K_{\text{off}} = \sum_i \sum_{m \neq 0} H_{ii}^{(m)}(l) |i\rangle \langle i| \otimes \hat{\sigma}_m + \sum_{i \neq j} \sum_m H_{ii}^{(m)}(l) |i\rangle \langle j| \otimes \hat{\sigma}_m. \quad (22)$$

This writing clarifies that in the Floquet case we have to deal with four kinds of matrix elements: diagonal in both Floquet and Hilbert space $H_{ii}^{(0)}(l)$, off-diagonal in Floquet space (i.e. $H_{ii}^{(m)}(l)$ with $m \neq 0$) but diagonal in Hilbert space, diagonal in Floquet space but off-diagonal in Hilbert space (i.e. $H_{ij}^{(0)}$ with $i \neq j$), and fully off-diagonal in both Floquet and Hilbert space ($H_{ij}^{(m)}(l)$ with $m \neq 0$ and $i \neq j$). By construction, only the first kind will remain at long flow time. In this notation, the Wegner-Floquet generator may be written:

$$\eta = [K_0, K_{off}] = \sum_{ij} \sum_n \left( H_{ij}^{(n)}(H_{ii}^{(0)} - H_{jj}^{(0)}) + \omega n H_{ij}^{(n)} \right) |i\rangle \langle j| \otimes \hat{\sigma}_n \quad (23)$$

$$\equiv \sum_{ij} \sum_n \eta_{ij}^{(n)} |i\rangle \langle j| \otimes \hat{\sigma}_n. \quad (24)$$

In the following, we shall focus on this Wegner-Floquet generator, as it is new to the literature and allows us to directly obtain the Floquet quasienergies with a single unitary transform. One feature that the Wegner-Floquet generator inherits from the static Wegner generator is that it works on the principle of energy-scale separation, and therefore is identically zero when applied to a translationally invariant system in real-space. Here, we will consider the case of disordered systems in real space where the on-site potential is strongly inhomogeneous, although we note that the method still works for clean systems, either with some small 'seed' randomness to break the homogeneity or when applied in momentum space. Before moving on to discuss the application of this generator, we first discuss the existence of conserved quantities of the Wegner-Floquet flow known as flow invariants.

### 4.3.1 Floquet Flow Invariants

An important property of a unitary flow in time-independent systems is the existence of flow invariants, conserved quantities of the exact flow-evolution [58]. These can be used to assess the validity of a given approximation scheme, as done in Refs. [61, 65], where the value of the invariant is compared at the start and end of the flow in order to verify the accuracy of the diagonalization process. In this section we establish an analogous flow invariant for the Wegner-Floquet flow discussed in Sec. 4.3, and in Sec. 6 we demonstrate a specific application.

In time-independent systems, the unitary flow of the Hamiltonian $H$ naturally leaves the eigenvalues unchanged, therefore it follows that there exist a family of flow invariants which may be defined as traces of integer powers of the Hamiltonian, $I_q(l) = \text{Tr}H^q(l)$ [58]. Our goal here is to obtain an equivalent measure for time-dependent systems. Importantly, the standard picture of flow invariants does not directly translate to time-dependent systems due to the time-dependence of the unitary transform and the tensor product structure of Eq. 19. The flow invariant we choose here is the simplest possible option, defined as the trace of $K$ in both real space and harmonic space, $I(l) = \text{Tr}[K(l)]$. The change of this quantity during the flow is given by:

$$\frac{\mathrm{d}}{\mathrm{d}l}\text{Tr}[K] = \text{Tr}\left[\sum_{ijk}\sum_{nm}\left(\eta_{ik}^{(n)}H_{kj}^{(m)} - H_{ik}^{(m)}\eta_{kj}^{(n)}\right)|i\rangle\langle j| \otimes \hat{\sigma}_{n+m} - \sum_{ij}\sum_{n}\omega n \eta_{ij}^{(n)}|i\rangle\langle j| \otimes \hat{\sigma}_n\right].$$

(25)

Plug in the explicit expression and $\eta_{ij}^{(n)}$ and take the trace:

$$\frac{\mathrm{d}}{\mathrm{d}l}\text{Tr}[K] = 2\sum_{ik}\sum_{n}H_{ik}^{(n)}H_{ki}^{(-n)}(H_{ii}^{(0)} - H_{kk}^{(0)}) = 0,$$

(26)

where the term with prefactor $\omega n$ vanishes due to the sum over $n \in [-N_h, N_h]$, and the remaining term vanishes due to the sum over all values of $(i, k)$. This quantity is therefore conserved by the flow, and by comparing this quantity computed at the start ($l = 0$) and end ($l \to \infty$) of the flow, we can verify the accuracy of our numerical procedure. We will come back to this Floquet flow invariant in Sec. 6 to assess the validity of our truncation scheme for weakly interacting systems.

## 5 Application I: The Driven Anderson Model

We shall first consider a prototype system of non-interacting fermions. It is worth emphasising up front that while for non-interacting systems the flow equation method may seem like an

elaborate way to solve a problem which can be more efficiently treated by exact diagonalization, the real advantage of this method lies in its ability to non-perturbatively solve *interacting* quantum systems on system sizes far larger than accessible to numerically exact methods. Our aim in the present section is to demonstrate the application of the flow equation method to a relatively simple non-interacting system where it can be applied (almost) exactly, and demonstrate that the results of the flow equation method agree well with the numerically exact solution. We shall then return to interacting systems in Section 6 after first demonstrating the method and establishing our notation.

## 5.1 The model

We now turn to a concrete example, namely a one-dimensional chain of length $L$ of non-interacting fermions in a disordered potential subject to periodic drive, i.e. a driven Anderson insulator [89–91]. The Hamiltonian is given by:

$$H(t) = F(t) \sum_{i=1}^{L} h_i c_i^\dagger c_i + G(t) \sum_{i=1}^{L-1} J_0 \left( c_i^\dagger c_{i+1} + c_{i+1}^\dagger c_i \right), \tag{27}$$

where the on-site energies are drawn from a box distribution $h_i \in [0, W]$, $J_0$ is the nearest-neighbour hopping amplitude and the functions $F(t)$ and $G(t)$ are some $T$-periodic functions which we leave as arbitrary for the moment.

We can expand these coefficients in terms of Fourier harmonics, which allows us to write down an ansatz in the full Floquet Hilbert space $\mathcal{F}$ for the form of the running (i.e. scale-dependent) Floquet operator $K(l)$. It takes the form:

$$K(l) = \sum_n \left( \sum_i h_i^{(n)} c_i^\dagger c_i + \sum_{ij} J_{ij}^{(n)} c_i^\dagger c_j \right) \otimes \hat{\sigma}_n + \mathbb{1} \otimes \omega \hat{n}, \tag{28}$$

with time-dependent coefficients determined by the relations $h_i(t, l) = h_i F(t) = \sum_n h_i^{(n)}(l) e^{in\omega t}$ and $J_{ij}(t, l) = J_{ij} G(t) = \sum_n J_{ij}^{(n)}(l) e^{in\omega t}$. We choose the time-independent coefficients $h_i^{(n)}$ and $J_{ij}^{(n)}$ to be real, and they satisfy the constraints $h_i^{(n)} = h_i^{(-n)}$ and $J_{ij}^{(n)} = J_{ji}^{(-n)}$ respectively. For non-interacting systems, this form of $K(l)$ turns out to be exact. Note that despite the initial Hamiltonian only containing nearest-neighbour hopping terms, in the running Floquet operator we must allow for arbitrarily long-range hopping terms $J_{ij}^{(n)} \forall i \neq j$ - one significant downside of Wegner-type generators is that they do not preserve the sparsity of matrices to which they are applied. In the present case, this results in the generation of long-range hopping terms at early stages of the flow, which decay to zero in the $l \to \infty$ limit but must nonetheless be kept track of, as well as the excitation of higher harmonics not present in the original microscopic model which likewise will decay at the end of the procedure.

## 5.2 Flow Equations

Following the Wegner-Floquet procedure outlined in Eq. 19, we can split the Floquet quasi-energy operator explicitly into diagonal and off-diagonal pieces as:

$$K_0 = \left[ \sum_i h_i^{(0)} c_i^\dagger c_i \right] \otimes \mathbb{1} + \mathbb{1} \otimes \omega \hat{n}, \tag{29}$$

$$K_{\text{off}} = \sum_{n \neq 0} \sum_i h_i^{(n)} c_i^\dagger c_i \otimes \hat{\sigma}_n + \sum_n \sum_{ij} J_{ij}^{(n)} c_i^\dagger c_j \otimes \hat{\sigma}_n, \tag{30}$$

where the superscripts are harmonic indices. From this, we can compute the generator $\eta = [K_0, K_{\text{off}}]$ which we can use to diagonalize the matrix $K$:

$$\eta = \sum_n \left( \sum_{ij} J_{ij}^{(n)} (h_i^{(0)} - h_j^{(0)}) c_i^\dagger c_j \right) \otimes \hat{\sigma}_n + \sum_{n \neq 0} \omega n \left( \sum_i h_i^{(n)} c_i^\dagger c_i + \sum_{ij} J_{ij}^{(n)} c_i^\dagger c_j \right) \otimes \hat{\sigma}_n. \quad (31)$$

The flow of the Floquet operator towards diagonal form is given by $dK/dl = [\eta, K]$. From this, following some algebra, we can obtain flow equations for the running couplings:

$$\frac{dh_i^{(n)}}{dl} = -n^2 \omega^2 h_i^{(n)} + 2 \sum_m \left[ \sum_j J_{ij}^{(n-m)} J_{ji}^{(m)} (h_i^{(0)} - h_j^{(0)}) \right]$$
$$+ \sum_m \omega(n-m) \sum_j \left( J_{ij}^{(n-m)} J_{ji}^{(m)} - J_{ij}^{(m)} J_{ji}^{(n-m)} \right), \quad (32)$$

$$\frac{dJ_{ij}^{(n)}}{dl} = -n\omega J_{ij}^{(n)} (h_i^{(0)} - h_j^{(0)} + n\omega) + \sum_m J_{ij}^{(n-m)} (h_i^{(0)} - h_j^{(0)})(h_j^{(m)} - h_i^{(m)})$$
$$+ \sum_m \sum_k J_{ik}^{(n-m)} J_{kj}^{(m)} (h_i^{(0)} + h_j^{(0)} - 2h_k^{(0)})$$
$$+ \sum_m \omega(n-m) \left( J_{ij}^{(m)} (h_i^{(n-m)} - h_j^{(n-m)}) + J_{ij}^{(n-m)} (h_j^{(m)} - h_i^{(m)}) \right)$$
$$+ \sum_m \omega(n-m) \sum_k \left( J_{ik}^{(n-m)} J_{kj}^{(m)} - J_{ik}^{(m)} J_{kj}^{(n-m)} \right), \quad (33)$$

which reduce back to the static flow equations for non-interacting fermions in the case where all coefficients with harmonic index $n \neq 0$ are zero (i.e. the time-independent case). Details of this calculation are shown in Appendix B.

The second term in Eq. 33 contains the familiar Wegner decay term $dJ_{ij}^{(n)}/dl \sim -J_{ij}^{(n)}(h_i^{(0)} - h_j^{(0)})^2$ seen in static systems (see Appendix A) which leads to the exponential decay of the off-diagonal elements during the flow. While the decay rules are modified in the time-dependent case, the key principle that Wegner-type transforms operate on is the idea of 'separation of energy scales', and this is apparent from the above flow equations: if the on-site energies are translationally invariant, the flow is identically zero. It follows from this that off-diagonal elements which couple sites very close in energy will decay exponentially slowly during the flow, requiring a long flow time in order for the system to be fully diagonalized. There is, however, no problem encountered due to resonant terms: for non-interacting systems, the sole effect of closely separated on-site energies is the need for a large flow time $l_{max}$.

Note that for discontinuous drive protocols, the expansion in terms of Fourier harmonics involves in principle infinitely many components, which is clearly impractical, so we must retain only a finite number of harmonics. In fact, even for continuous drive protocols, the equations above are not closed for a finite number of harmonics, meaning that we must make a suitable truncation and choose to keep only $N_h$ harmonics (with the harmonic index $n \in [-N_h, N_h]$). The choice of a suitable value for $N_h$ will depend on the problem in question. Careful inspection of the above equations reveals that for large harmonic index $n$ the equations reduce to $dh_i^{(n)}/dl \approx -n^2 \omega^2 h_i^{(n)}$ and $dJ_{ij}^{(n)}/dl \approx -n^2 \omega^2 J_{ij}^{(n)}$ and so these coefficients will decay exponentially in flow time for large values of $n$. This means that terms with large harmonic indices typically do not play a significant role except at very low frequencies, and only the lowest-order harmonics in the Fourier expansion need be retained in order for the procedure to be accurate. Bearing in mind that not all couplings are independent (e.g. $J_{ij}^{(n)} = J_{ji}^{(-n)}$), in total this results in $(2N_h + 1) \times L(L+1)/2$ coupled ordinary differential equations (where $L$ is the number of real-space sites), which can be straightforwardly solved numerically.

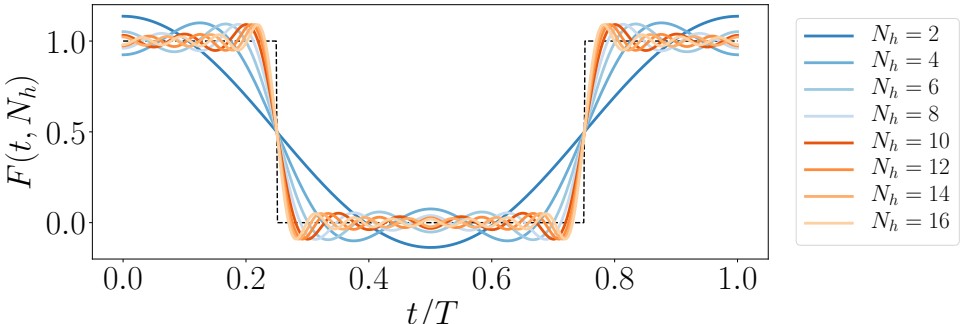

Figure 2: A comparison of the drive for varying $N_h$ showing how the harmonic expansion approximates the discontinuous form of the drive as $N_h$ is increased. The discontinuous step drive ($N_h \to \infty$) approximated by $F(t, N_h)$ is indicated by the black dashed line.

An interesting feature of the above flow equations is that in the limit of very high frequency $\omega \to \infty$, the harmonics with $n > 0$ again reduce to $dh_i^{(n)}/dl \approx -n^2\omega^2 h_i^{(n)}$ and $dJ_{ij}^{(n)}/dl \approx -n^2\omega^2 J_{ij}^{(n)}$ and decay exponentially to zero, with essentially zero feedback on the $n = 0$ harmonics. In other words, it is explicit from the above equations that at high frequencies the problem reduces to a static problem with the couplings given by their time-averaged ($n = 0$) values, as one might reasonably expect. Also, if we take the case of $J_{ij} = 0 \quad \forall \quad i, j$ then the flow equations return the time-averaged $h_i$, i.e. just the zeroth harmonic $h_i^{(0)}$: all the harmonics $h^{(n)}$ for $n > 0$ are decoupled from the $h^{(0)}$ term and decay exponentially to zero.

## 5.3 Floquet quasienergies

For the following results, we use a discontinuous step-like drive, as this represents the most challenging case for the harmonic expansion we employ. The Hamiltonian we consider is:

$$H(t) = \begin{cases} \sum_{i=1}^{L} h_i c_i^\dagger c_i, & \text{if } T/4 \leq t < 3T/4 \\ \sum_{i=1}^{L-1} J_0\left(c_i^\dagger c_{i+1} + c_{i+1}^\dagger c_i\right), & \text{otherwise} \end{cases}, \tag{34}$$

where the function $F(t)$ is now chosen to be a step function to match the above Hamiltonian, with $G(t) = 1 - F(t)$ in Eq. 27. Note that this is a rather different choice from the monotonic drive studied in many other works [89–91]: the step-like drive consists of alternately applying the diagonal and off-diagonal parts of the Hamiltonian, which at high frequencies yields a (time-averaged) Anderson Hamiltonian, but at low frequencies can exhibit richer behaviour. We discuss the low-frequency physics of this system further in Appendix C: here, we will simply use this model to demonstrate that flow equations agree with numerically exact methods.

We will compute the Fourier harmonics of this drive (which has been chosen to have purely real Fourier components), and keep $N_h$ of the harmonics for the flow equation procedure, such that the driving protocol is described by a function $F(t, N_h)$. Our purpose in choosing a step drive is partly because this is a common choice in the simulation of driven systems (see, e.g., Refs. [32,45]), and partly because it represents a worst-case scenario in terms of computational complexity for the method, as formally we require infinite many Fourier components to exactly realise the discontinuous form of the drive. The form of $F(t, N_h)$ is shown in Fig. 2, which demonstrates how well the step drive is approximated as a function of the number of harmonics $N_h$.

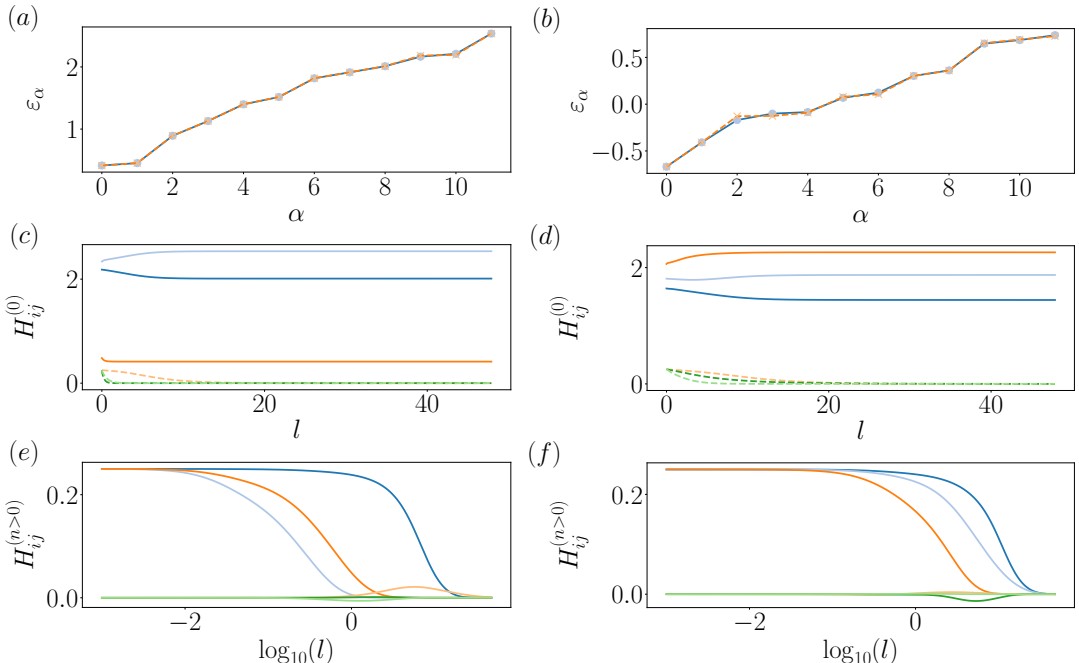

Figure 3: Two representative samples of the quasienergies and flow of the couplings for two different disorder realisations at different drive frequencies (left and right columns respectively), with $L = 12$, $W = 5$, $J_0 = 0.5$ and $N_h = 5$ harmonics. The left column shows the results for a dimensionless drive frequency $\omega/W = 2\pi/5 \approx 1.26$ which is larger than the disorder bandwidth $W$, while the right column shows results for $\omega/W = 2\pi/20 \approx 0.31$ which is much smaller than the disorder bandwidth. Panels (a) and (b) show the quasienergies $\varepsilon_n$ obtained by the FE method (orange) compared with the results from ED (blue). The higher frequency result is more accurate; more harmonics are required at lower frequencies. Panels (c) and (d) show the flow of a representative subset of the zero-frequency terms - the solid lines are the $h_i^{(0)}$ coefficients, while the dashed lines are the $J_{ij}^{(0)}$. Panels (e) and (f) show the flow of a representative subset of finite-frequency harmonics, which decay to zero extremely quickly at high drive frequencies (note the logarithmic scale) but much more slowly for small $\omega$.

In Fig. 3 we show the quasienergies and the flow of the couplings at two different drive frequencies for system size $L = 12$, disorder strength $h_i \in [0, W]$ with $W = 5$ and hopping $J_0 = 0.5$. We compare the quasienergies obtained with the flow equation (FE) approach with those computed using exact diagonalization (ED) using the QuSpin package [92,93]. In the ED results, we use the exact Hamiltonian given by Eq. 34, i.e. we implement the step drive exactly with no approximation. For illustrative purposes we keep the system size small and only show the behaviour of a small but representative sample of parameters, to prevent the plots from becoming cluttered with too many couplings, and we show the flow up to $l_{max} = 50$, though typically we use a larger $l_{max} \sim 10^3$ to ensure convergence. To reduce the numerical load, we set all off-diagonal couplings to zero when they decay below $10^{-6}$. In principle, to reduce the computational cost further, one could dynamically reduce the number of equations when the off-diagonal couplings have decayed, however we did not find it necessary to implement this feature here.

In Fig. 4a we demonstrate the accuracy versus number of retained harmonics $N_h$ for a

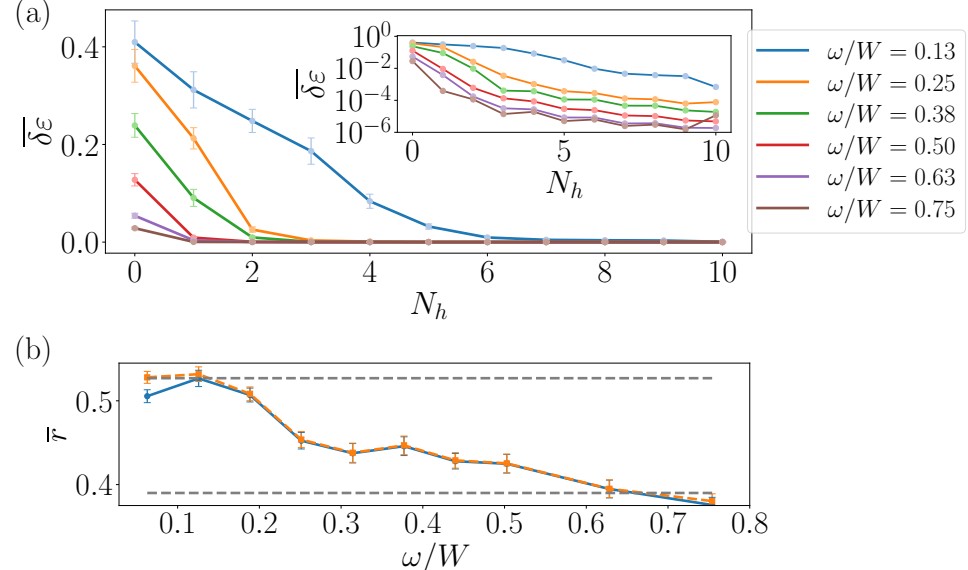

Figure 4: (a) Relative error in the quasienergies (as compared with exact diagonalization) with $N_h$ for $L = 12$, disorder strength $W = 5$ and hopping $J_0 = 0.5$ for several different frequencies. The results were averaged over 128 disorder realisations, and the inset shows the same data on a logarithmic scale. Error bars represent the variance over disorder realisations. For lower drive frequencies, a larger number of harmonics must be retained in order to obtain accurate quasienergies. This follows intuitively from the idea that high frequency drive can be described by the time-averaged behavior, i.e. the zeroth harmonic, and so the higher the drive frequency, the fewer harmonics are required. In order to show the effect of retaining a large number of harmonics, we keep the system size small. (b) Level spacing statistics computed with flow equations (blue) and exact diagonalization (orange) for the same parameters, with $N_h = 10$ harmonics. The grey dashed lines represent $\overline{r} \approx 0.39$ and $\overline{r} \approx 0.53$ corresponding to Poisson and Circular Ensemble [25, 26] level statistics respectively. Note that at the lowest frequency shown here, there starts to be a deviation between ED and FE results: for very low frequencies, we require a larger value of $N_h$ in order to obtain accurate results.

larger range of frequencies and system sizes by plotting the relative error, defined as:

$$\delta\varepsilon = \frac{1}{L} \sum_{\alpha} \left| \frac{\varepsilon_{\alpha}^{ED} - \varepsilon_{\alpha}^{FE}}{\varepsilon_{\alpha}^{ED}} \right|, \tag{35}$$

where $\varepsilon_{\alpha}^{ED/FE}$ are the eigenvalues obtained with exact diagonalization (ED) and flow equations (FE) respectively. This is then averaged over $N_s = 128$ disorder realisations. For sufficiently large values of $N_h$, we find excellent agreement between the FE and ED results at all frequencies. Additionally, we find that the flow invariant $I(l)$ defined in Sec. 4.3.1 is conserved throughout the flow to high precision for all frequencies considered here. Numerically, we find that the disorder averaged change in this quantity over the course of the flow is extremely small, with $\overline{\delta I} = |I(l=0) - I(l \to \infty)| \lesssim 10^{-12}$.

As a further, far more demanding check of our method, we also compute the level spacing statistics as a function of frequency, a metric which has been extensively used in studies of many-body localization to distinguish localized and delocalized phases [94]. Specifically, we

compute the following quantities:

$$\delta_\alpha = |\varepsilon_\alpha - \varepsilon_{\alpha+1}|, \tag{36}$$

$$r_\alpha = \min(\delta_\alpha, \delta_{\alpha+1})/\max(\delta_\alpha, \delta_{\alpha+1}), \tag{37}$$

and we compute $r = \overline{\langle r_n \rangle}$, where the $\langle ... \rangle$ represents the average within a sample and the overline the average over disorder realisations. In a localized phase, the energy level spacings will be distributed according to a Poisson distribution $P(\delta) = \exp(-\delta)$, corresponding to $r \approx 0.39$. In a delocalized phase, however, the energy levels will exhibit level repulsion: while for undriven systems in a delocalized phase the spacings will follow the Wigner-Dyson distribution, which may be approximated by the Wigner surmise $P(\delta) = (\pi/2)\delta\exp(-\pi\delta^2/4)$ (leading to $r \approx 0.53$), for *driven* systems where the Floquet eigenvalues lie in the folded range $[-\omega/2, \omega/2]$, the correct ensemble is the Circular Ensemble (CE) [25, 26]. Numerically, this also gives $r \approx 0.53$ in the delocalized phase. The results for a system size $L = 12$ with $N_h = 10$ harmonics are shown in Fig. 4b, and demonstrate the precise agreement between flow equation and exact diagonalization results even for a numerically highly demanding quantity, for all except the very lowest frequency considered where a larger value of $N_h$ is required. The behavior of the average level spacing shown in Fig. 4b indicates a crossover from localized to delocalized behavior as a function of the drive frequency. As shown in Appendix C, this is a finite size effect and the delocalized region shrinks as the system size is increased. Interestingly as we show in Appendix C this crossover is absent for a monochromatic drive, where the level spacing remains close to the localized value for all frequencies and depends weakly on system sizes.

## 5.4 Floquet Eigenstates

In addition to the quasienergies, in order to have a complete solution to the problem one must compute the Floquet eigenstates. For a non-interacting system, in the $l = \infty$ basis where the Floquet Hamiltonian $\tilde{H}_F$ is diagonal, the eigenstates are the single-particle states of $\tilde{H}_F$, given by $|\psi_i\rangle = c_i^\dagger(l = \infty)|0\rangle$ for $i \in [1, L]$. For static systems (see Appendix A.2), in order to obtain the eigenstates we simply need to invert the unitary transform to transform $c_i^\dagger(l = \infty)$ *backwards* from $l = \infty$ to $l = 0$. An additional complication in the case of Floquet systems is that if we want to compute the Floquet eigenstates, then we do not want to obtain the eigenstates in the initial microscopic basis, where they will be time-dependent. Strictly speaking, obtaining the Floquet eigenstates is a two-step process: the first step is the inversion of the Wegner-Floquet unitary transform to obtain time-dependent eigenstates in the initial basis, followed by a Type II unitary transform (see Sec. 4.2) to get rid of the time-dependence. We can nonetheless approximate the result of this two-step procedure with a single quasi-unitary transform by reversing only the flow of the zero-frequency terms in the Hamiltonian. We make the following ansatz for the flow of the creation operator:

$$c_i^\dagger(l) = \sum_j \beta_{i,j}^{(0)}(l)c_j^\dagger, \tag{38}$$

with initial condition $\beta_{i,j}^{(0)}(l = \infty) = \delta_{i,j}$ and we obtain the flow equation for this operator by considering only the zero frequency components of the generator:

$$\frac{dc_i^\dagger(l)}{dl} = \sum_j \beta_{i,j}^{(0)}(l)[\eta^{(0)}, c_j^\dagger] = \sum_{jk} J_{jk}^{(0)}(h_j - h_k)\beta_{i,k}^{(0)}c_j^\dagger, \tag{39}$$

giving a flow equation for the coefficients $\beta_{i,j}$

$$\beta_{i,j} = \sum_k J_{jk}^{(0)}(h_j - h_k)\beta_{i,k}^{(0)}, \tag{40}$$

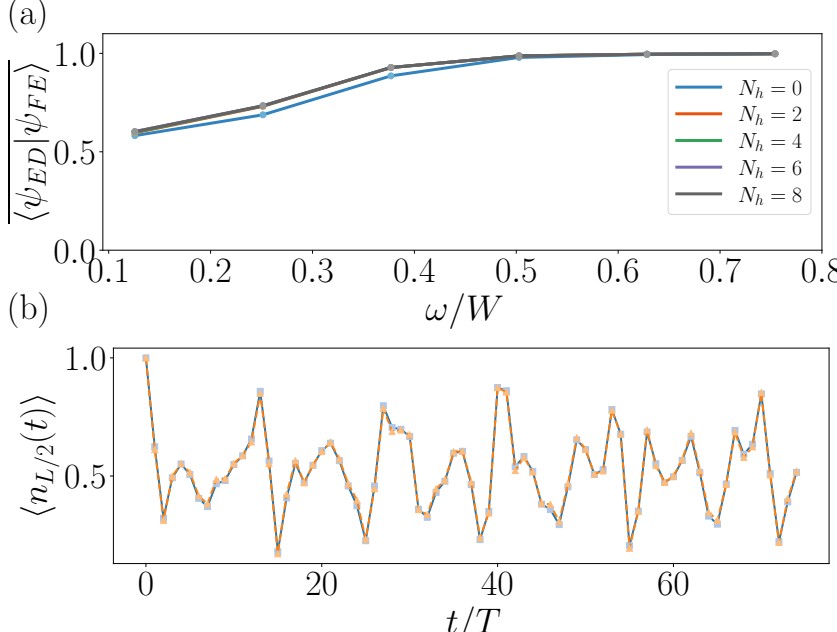

Figure 5: (a) Overlap between the Floquet eigenstates obtained with exact diagonalization and approximate states obtained with the flow equation method, again with $L = 12$, $W = 5$ and $J_0 = 0.5$ and averaged over $N_s = 128$ disorder realisations. Error bars representing the variance over the disorder realisations are smaller than the plot markers. As described in the text, the approximation used here is better at higher frequencies. Note that there is very little dependence on harmonic number $N_h$, with the lines for $N_h \geq 2$ all lying atop one another, as Eq. 38 takes into account non-zero harmonics only implicitly. (b) A representative example of the stroboscopic dynamics for a system of size $L = 12$ with $N_h = 8$ harmonics at a dimensionless frequency $\omega/W = \pi/5 \approx 0.63$. The flow equation result is shown in blue (solid line), with the result from exact diagonalization in orange (dashed) : the agreement is excellent, and the lines are almost precisely on top of one another.

which we can use to approximately construct the Floquet eigenstates. This procedure works best at high frequency, however for all but the lowest frequencies the results are of reasonable accuracy, with an overlap of close to unity for frequencies $\omega/W > \pi/10$ for the parameters considered here. (At low frequencies, this inversion of the flow for the $n = 0$ terms deviates strongly from unitarity, hence the larger error.)

In Fig. 5a, we show the overlap between the eigenstates generated by the flow equation procedure and the output of an exact diagonalization calculation. At high drive frequencies, this approximate method is able to reproduce the eigenstates with extremely high fidelity (up to an unimportant arbitrary phase factor), however at frequencies much lower than the disorder bandwidth the approximation fails and the full unitary transform must be used. We emphasise that while this can be done exactly as a two-step process using two different generators, as explained above, we do not explore this option here, as in this work we are more interested in exploring what may be obtained from a single unitary transform using the Wegner-Floquet generator.

## 5.5 Stroboscopic Dynamics

Any initial state can be decomposed in terms of Floquet eigenstates, and with the complete knowledge of the eigenstates and eigenvalues, we can perform this decomposition and time-evolve any initial state we choose. Specifically, an arbitrary state $|\phi(t)\rangle$ at a time $t = 0$ can be expressed in terms of the Floquet eigenstates $|\psi_\alpha(t)\rangle$ like so:

$$|\phi(t=0)\rangle = \sum_\alpha |\psi_\alpha(t=0)\rangle \langle \psi_\alpha(t=0)|\phi(t=0)\rangle = \sum_\alpha c_\alpha |\psi_\alpha(t=0)\rangle , \qquad (41)$$

with $c_\alpha = \langle \psi_\alpha(t=0)|\phi(t=0)\rangle$. The time evolution of this state for some time $t > 0$ is given by:

$$|\phi(t)\rangle = \sum_\alpha c_\alpha e^{-i\varepsilon_\alpha t} |\psi_\alpha(0)\rangle , \qquad (42)$$

where $t$ is an integer multiple of the drive period $T$. This is known as stroboscopic evolution. As an example, the expectation value of the density on site $i$ is given by:

$$\langle n_i(t)\rangle = \langle \phi(t)|n_i|\phi(t)\rangle = \sum_{\alpha\beta} c_\alpha^* c_\beta \exp[i(\varepsilon_\alpha - \varepsilon_\beta)t] \langle \psi_\alpha(0)|n_i|\psi_\beta(0)\rangle . \qquad (43)$$

The results of the stroboscopic evolution from an initial charge density wave state $|1010...\rangle$ are shown in Fig. 5b. The agreement between exact diagonalization and flow equation results is almost perfect. As the system we consider here is a free fermion Anderson insulator which does not exhibit dephasing, the late time states do not synchronise with the drive [45], however we nonetheless show the results of this calculation as a proof of principle that this method allows one to recover the full dynamical behavior in addition to the quasienergies.

# 6 Application II: Weakly Interacting Disordered Fermions

## 6.1 The model

Now that we have covered driven non-interacting systems and demonstrated how to completely solve them using the flow equation approach, let us discuss how to add interactions into the picture. We now start from a time-dependent, *interacting* fermionic Hamiltonian of the form:

$$H = \sum_i h_i(t)c_i^\dagger c_i + \sum_{ij} J_{ij}(t)c_i^\dagger c_j + \frac{1}{2} \sum_{ij} \Delta_{ij}(t)c_i^\dagger c_j^\dagger c_j c_i , \qquad (44)$$

where the coefficients $h_i(t)$, $J_{ij}(t)$ and $\Delta_{ij}(t)$ all have some arbitrary (periodic) time-dependence, and here we allow for arbitrary long-range couplings from the start. The static form of this Hamiltonian has been studied using flow equation methods in the context of many-body localization (MBL) in Ref. [61] in the case of short-range couplings and in Ref. [65] in the case of long-range couplings. In the static case, the diagonal Hamiltonian can be written as a series of mutually commuting $n$-body interaction terms, $\check{H} = \sum_i h_i n_i + \frac{1}{2} \sum_{ij} \tilde{\Delta}_{ij} n_i n_j + \sum_{ijk} \Gamma_{ijk} n_i n_j n_k + \ldots$, which requires us to keep track of a prohibitively large number of terms during the diagonalization process for any system of realistic size. Fortunately, in certain cases, one can perform a truncation of this series and keep only the lowest-order terms. Following the ansatz described in Refs. [61, 65], we will here assume a scale-dependent Floquet operator of the form:

$$K(l) = \sum_n \left[ \sum_i h_i^{(n)} c_i^\dagger c_i + \sum_{ij} J_{ij}^{(n)} c_i^\dagger c_j + \frac{1}{2} \sum_{ij} \Delta_{ij}^{(n)} c_i^\dagger c_j^\dagger c_j c_i \right] \otimes \hat{\sigma}_n + \mathbb{1} \otimes \omega \hat{n} , \qquad (45)$$

and discard all newly-generated terms outside of this manifold. For the static system, this ansatz is valid either in the strongly disordered, many-body localized regime [61] or in the weakly-interacting regime (where the generation of higher-order terms is suppressed by powers of the interaction strength, which we choose here to be small - see Ref. [65] for details regarding the generation of higher order terms due to the interactions). Here, we will assume weak interactions[2] and work in the regime where the static Hamiltonian would be in the many-body localized phase. Note that this truncated form of $K(l)$ is not the result of a perturbative analysis, but instead arises from a truncation in operator space which is valid in the localized phase. Flow equation methods based around Wegner-type generators therefore avoid problems with resonances [60] which can otherwise lead to divergent terms in perturbative treatments of interacting systems. By analogy with the non-interacting system in Section 5, we may expect to see a delocalization crossover or transition as a function of the drive frequency, at which point this ansatz for the form of $K(l)$ will break down; we will examine the evidence for this later. Accurately capturing the delocalized phase requires an ansatz that goes beyond form form shown in Eq. 45, including additional terms which must be kept track of during the flow. The final fixed-point diagonal Floquet operator will be of the form $\tilde{K} = \tilde{H} \otimes \mathbb{1} + \mathbb{1} \otimes \omega \hat{n}$, with $\tilde{H}$ given by:

$$\tilde{H} = \sum_i \tilde{h}_i n_i + \frac{1}{2} \sum_{ij} \tilde{\Delta}_{ij} n_i n_j \,, \tag{46}$$

from which we can construct the many-body quasienergies simply by applying this Hamiltonian to each of the $2^L$ many-body states.

## 6.2 Flow equations

The Floquet quasienergy operator may again be separated into diagonal and off-diagonal parts as $K = K_0 + K_{\text{off}}$, now with:

$$K_0 = \left[ \sum_i h_i^{(0)} c_i^\dagger c_i \right] \otimes \mathbb{1} + \mathbb{1} \otimes \omega \hat{n} + \frac{1}{2} \left[ \sum_{ij} \Delta_{ij}^{(0)} c_i^\dagger c_j^\dagger c_j c_i \right] \otimes \mathbb{1} \,, \tag{47}$$

$$K_{\text{off}} = \left[ \sum_{n \neq 0} \sum_i h_i^{(n)} c_i^\dagger c_i \otimes \hat{\sigma}_n + \sum_n \sum_{ij} J_{ij}^{(n)} c_i^\dagger c_j \otimes \hat{\sigma}_n \right] + \frac{1}{2} \sum_{n \neq 0} \left[ \sum_{ij} \Delta_{ij}^{(n)} c_i^\dagger c_j^\dagger c_j c_i \right] \otimes \hat{\sigma}_{\mathfrak{m}} \,. \tag{48}$$

The calculation proceeds similarly to the non-interacting case, but now the generator acquires additional terms due to the interactions. These new contributions lead to the flow equation for the interaction term:

$$\frac{\mathrm{d}\Delta_{ij}^{(n)}}{\mathrm{d}l} = -\omega^2 n^2 \Delta_{ij}^{(n)} + 2 \sum_m \sum_{k \neq i,j} \left[ J_{ik}^{(n-m)} J_{ik}^{(m)} (\Delta_{ij}^{(0)} - \Delta_{kj}^{(0)}) + J_{jk}^{(n-m)} J_{jk}^m (\Delta_{ij}^{(0)} - \Delta_{ik}^{(0)}) \right] \,, \tag{49}$$

where we refer the reader to Refs. [61,65] and Appendix B for details on how to compute the flow of the interacting terms. For the interacting system within the approximation of Eq. 45, there are now $L^2 + N_h \times L(3L+1)/2$ coupled differential equations to solve. Note that the total number of equations scales polynomially in the system size (and linearly in $N_h$), in contrast with exact diagonalization where the size of the Hilbert space scales exponentially with the

---

[2]By comparison with previous work [61, 62, 65], here we use the convention where the normal-ordering corrections [95] are computed with respect to the vacuum and evaluate to zero. We will address the effect of stronger interactions and non-zero normal-ordering corrections in more detail in a future work.

system size. In regimes where this approximation can be applied, flow equations can therefore access far larger system sizes than numerically exact methods are capable of reaching. We first demonstrate that the approximation of Eq. 45 is accurate before providing a proof-of-concept demonstration of a quantity that cannot be computed with any other method.

## 6.3 Quasienergies

We again compare the eigenvalues obtained using the FE method with the results of an exact diagonalization calculation. Here we use the same driving protocol as in Section 5, and choose the interaction term to be a time-independent, nearest-neighbour coupling satisfying $\Delta_{ij}(t) = \delta_{i,i\pm1}\Delta_0$. Fig. 6 shows the results for $L = 5$, $N_h = 3$, $\omega = 2\pi$, $d \in [0, W]$ with $W = 5$ and $\Delta_0 = 0.01$. Again, we keep the system size small for illustrative purposes, however we will show results for larger systems in Section 6.5. We find that in the weakly interacting regime and at high frequencies, the agreement is excellent.

The accuracy becomes significantly worse at low frequencies, more so than in the non-interacting system. In that case, there was a crossover to a delocalized state as a function of frequency. It is reasonable to assume that the interacting system will also exhibit some form of delocalization as the driving frequency is lowered, and at this point the ansatz (Eq. 45) used for the many-body Hamiltonian will break down. This is confirmed by Fig. 7, in which we show the median relative error versus frequency for three different values of $N_h$. All three curves shown in Fig. 7 are essentially the same, with only minor variations, confirming that the dominant error in the interacting system is not the choice of harmonic number $N_h$, but instead the breakdown of the ansatz Eq. 45 at low frequencies. In Sec. 6.5, we shall examine this in more detail from the perspective of local integrals of motion, but first we shall introduce a second self-consistent error estimation method that does not rely on a comparison with exact numerics.

## 6.4 Error Estimation based on Flow Invariant

We have shown that the quasienergies obtained using flow equations match those obtained by exact diagonalization for small systems, however if we wish to use the flow equation method to study systems larger than those accessible to numerically exact methods, we require a secondary error estimation method. To this extent we take advantage of the concept of flow invariant, which we have introduced for the Wegner-Floquet flow in Sect. 4.3.1, and which provide an internally self-consistent check of the accuracy. Following the discussion in Sec. 4.3.1 we now introduce a flow invariant for the running Floquet operator in Eq. (45) that we can write as

$$I(l) = 2\sum_i h_i^{(0)}(l) + \sum_{i\neq j} \Delta_{ij}^{(0)}(l),$$

where here the trace is taken over all two-particle states, the simplest non-trivial choice which remains practical even for large system sizes. In the case under consideration, the expression for the invariant can be further simplified by noticing that, in absence of normal ordering corrections [61], the flow of the interaction terms $\Delta_{ij}$ is decoupled from the flow of the quadratic terms $h_i$ and $J_{ij}$ which retains the same form as for the non-interacting driven problem. As discussed in Sec. 5, the flow invariant for the quadratic problem is conserved. We can therefore define a suitable flow invariant for the many-body problem by looking at the interaction terms only, and considering their sum total at the start and end of the flow. As newly-generated interaction terms may be negative, the sum $\sum_{i\neq j} \Delta_{ij}^{(0)}(l)$ is typically conserved throughout the flow, with divergent positive terms being cancelled by equally divergent negative terms. We

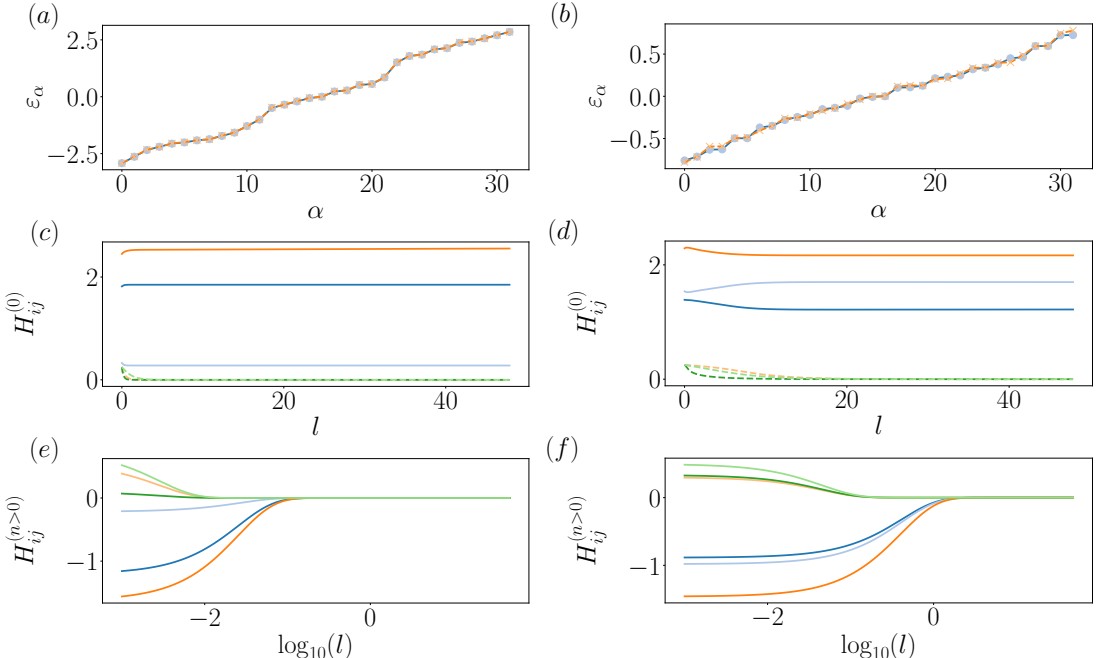

Figure 6: A comparison of the results for weakly-interacting systems at two different dimensionless frequencies, $\omega/W = 2\pi/5$ (left column) and $\omega/W = 2\pi/20$ (right) with $N_h = 5$, $L = 5$, $W = 5$, $J_0 = 0.5$ and interaction strength $\Delta_0 = 0.01$. As in Fig. 3, panels (a) and (b) show the quasienergies obtained with exact diagonalization and flow equation methods, while panels (c) and (d) show the flow of a representative sample of zero-frequency terms and panels (e) and (f) show the flow of a representative sample of terms with $n > 0$. The agreement between FE and ED quasienergies starts to visibly deviate at lower frequencies. We conjecture that this is due to the proximity to a phase transition where the ansatz of Eq. 45 breaks down; in Section 6.5 we explore this in further detail.

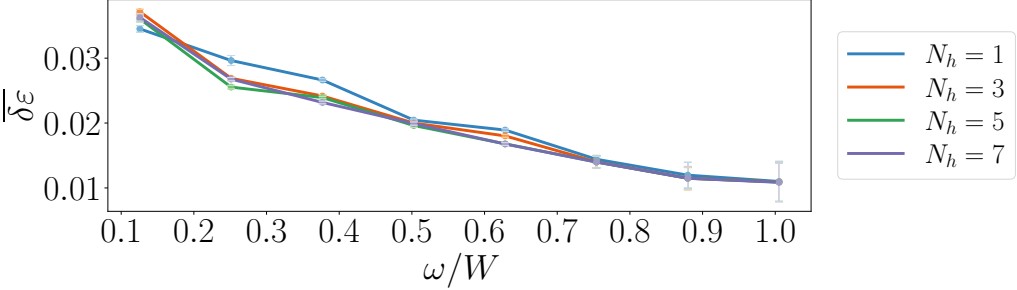

Figure 7: The relative error of the quasienergies obtained by the FE method for an interacting system of size $L = 12$ and disorder strength $W = 5$, averaged over $N_s = 64$ samples. Error bars indicate the variance over disorder realisations. Here, we restrict ourselves to the half-filled eigenstates. We find that the accuracy is almost independent of the choice of $N_h$, and conclude that the main source of error in the interacting system is the breakdown of the anastz (Eq. 45) used for the running Hamiltonian at low frequencies.

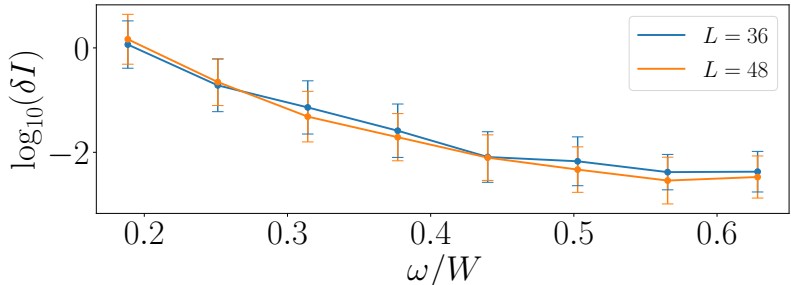

Figure 8: Conservation of the flow invariant for two system sizes, $L = 36$ (blue) and $L = 48$ (orange), with disorder strength $W = 5$, interaction strength $\Delta_0 = 0.01$, hopping $J_0 = 0.5$ and retaining $N_h = 3$ harmonics. The results were averaged over between $N_s = 64 - 256$ disorder realisations: here we show the median, and the error bars are given by the median absolute deviation.

therefore introduce our flow invariant as the sum of the modulus of the interaction terms:

$$\mathcal{I}(l) = \frac{1}{L^2} \sum_{i \neq j} |\Delta_{ij}^{(0)}(l)|, \tag{50}$$

which we normalize by $1/L^2$ for convenience [58]. We are interested in the quantity:

$$\delta \mathcal{I} = |\mathcal{I}(l \to \infty) - \mathcal{I}(l = 0)|, \tag{51}$$

which measures deviations from unitarity. While the sum of the *modulus* of the interaction terms is not strictly conserved, it nonetheless serves as a reliable indicator of divergent terms. This is very similar in spirit to the quantities studied in Refs. [58, 61, 65].

The results are shown in Fig. 8 for the same two system sizes that will be considered in Sec. 6.5, averaged over $N_s = 64 - 256$ disorder realisations. We find that at the lowest frequencies considered, there is significant deviation from unitarity, which implies that it is necessary to include additional terms in Eq. 45 in order to accurately capture the behaviour of the system. As in Refs. [61, 65], this likely signifies a transition into a delocalized phase. At higher frequencies, however, the flow invariant is very well conserved, implying that the system is strongly localized. We now explore this point further from the point of view of emergent integrals of motion, quantities typically used to characterize many-body localization in time-independent systems.

## 6.5 Floquet Integrals of Motion

The final fixed-point diagonal Hamiltonian for the weakly-interacting problem, as shown in Eq. 46, now contains only mutually commuting terms and is given by $\tilde{H} = \sum_i \tilde{h}_i n_i + \frac{1}{2} \sum_{ij} \tilde{\Delta}_{ij} n_i n_j$. In previous works on static MBL [60, 61, 65, 96, 97], the real-space decay of the $\tilde{\Delta}_{ij}$ terms has been associated with the dephasing length of the local integrals of motion (or $l$-bits) that characterise MBL matter. These terms are expected to decay exponentially as a function of distance for localized systems with short-range couplings [61, 96, 97] and may exhibit more exotic behavior for systems with long-range couplings [65]. In presence of periodic drive one would expect these $l$-bits to remain well defined at least for sufficiently high driving frequency [27]. Here we explicitly construct the Floquet $l$-bits and study their interaction as a function of frequency.

The results are shown in Fig. 9 for two system sizes, $L = 36$ and $L = 48$, retaining $N_h = 3$ harmonics and averaged over $N_s = 64$ or $N_s = 128$ disorder realisations for $L = 36$ and $L = 48$

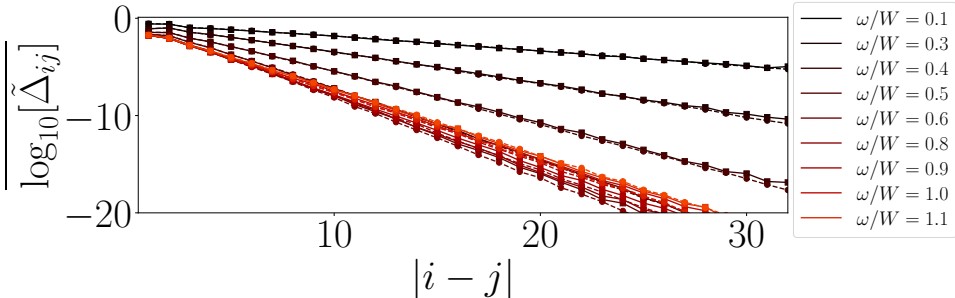

Figure 9: Decay of the Floquet $l$-bit couplings with distance for several different driving frequencies $\omega$, and system sizes $L = 36$ (solid lines with square markers) and $L = 48$ (dashed lines with circular markers) , $N_h = 3$. $W = 5$, $J_0 = 0.5$ and $\Delta_0 = 0.01$ both averaged over $N_s = 256$ disorder realisations. Independently of system size, there appears to be a localization/delocalization transition as a function of the driving frequency $\omega$, visible by the change of the decay of the Floquet $l$-bits from an approximately frequency-independent exponential decay to a much slower exponential decay below $\omega_c/W \approx 0.5$. Note that at the lowest frequencies shown, the results must be considered qualitative only.

respectively. We find that at high drive frequencies, these 'Floquet integrals of motion' (or Floquet $l$-bits) decay exponentially with distance as in the conventional static many-body localized system, however at low drive frequencies they appear to flatten out, consistent with the system becoming less localized and potentially undergoing a transition to a delocalized phase. We caution that our results in the low frequency regime are qualitative only: in the absence of normal-ordering corrections [56,61,65,95] Eq. 49 can exhibit an unphysical exponential growth in flow time $l$ in the delocalized regime for small $|i-j|$, visible in Fig. 9 at the smallest driving frequencies and at short distances. This is unrelated to our choice of $N_h$ and as such retaining more harmonics would not improve the accuracy in this region. Our aim in this work was to present the simplest possible form of the flow for the driven interacting system as a proof-of-concept: we will explore the consequences of including these normal-ordering corrections in a future work dedicated to interacting systems.

## 7 Conclusion

In conclusion, we have demonstrated a new method for the diagonalization of Floquet quasienegy/evolution operators based around continuous unitary transforms, or 'flow equations'. This method is a generalisation of the Wegner flow which has been used in a variety of contexts in the years since its development [56,74]. Here we make use of the Floquet theorem to rewrite a periodically driven system as a time-independent system in a larger composite Hilbert space amenable to treatment with flow equation techniques. We have briefly reviewed a variety of different choices of generator and shown where our choice fits in, and we have given several examples of both non-interacting and weakly interacting quantum systems where this method is capable of reproducing exact numerical results as well as going beyond the state-of-the-art accessible to other methods, which we have demonstrated by computing for the first time the 'Floquet $l$-bits' of a driven many-body localized system on length scales inaccessible to all other methods.

The strengths of this method go beyond what we have shown here, however, and we close

this article with a look to the future. Flow equations have previously been used to study many-body localization in two-dimensional systems [61] as well as in coupled chains [62], and a generalisation of the results shown in Section 6 to two-dimensional driven systems is straightforward. Equally, flow equation methods are not limited to the models considered here: one particularly intriguing application is the study of time crystal behavior in driven transverse field Ising models in one and two dimensions, something which we will address separately in a forthcoming work. Another clear avenue for further improvement is the treatment of more strongly interacting quantum systems, through the incorporation of advanced non-perturbative techniques [56, 61, 65, 95] and an improved ansatz for the running Hamiltonian in the 'Floquet-MBL' phase, perhaps incorporating higher-order terms in the operator expansion beyond the truncation considered here. As our method lends itself naturally to continuous drive protocols, as well as to discontinuous drives when expanded in terms of harmonics, it may also be interesting to use Floquet flow equations to examine continuous drives in more detail: this is a particular strength of the method, as having to retain a large number of harmonics in order to capture a discontinuous drive imposes a severe limit on the system sizes achievable. By considering smooth drive protocols and retaining only a small number of harmonics, even larger system sizes are available to this method and will be explored in other works in the near future.

## Acknowledgements

All numerical computations were performed on the Collège de France IPH computer cluster. We acknowledge use of the QuSpin exact diagonalization library for benchmarking our flow equation code [92, 93].

**Author contributions**    The central formalism in Section 4 was developed by DM under the supervision of SJT and MS. The results in Sections 5 and 6 were obtained by SJT. The project was conceived by and directed by MS. All authors contributed towards the writing of the final manuscript.

**Funding information**    SJT acknowledges financial support from the DIM SIRTEQ grant DynDisQ. DM acknowledges support from Fundação para a Ciência e a Tecnologia (Portugal) through project UIDB/EEA/50008/2020 and partial support from LabEx ENS-ICFP (ANR-10-IDEX-0001-02). MS acknowledges financial support from the ANR grant "NonEQuMat" (ANR-19-CE47-0001).

## A   Example: Flow Equations for Static Hamiltonians

In this Appendix, we sketch the application of the flow equation technique to a static, non-interacting Hamiltonian to illustrate the exact application of the method for readers unfamiliar with the technique. Additionally, we show for the first time how to obtain the eigenstates of the non-interacting system using flow equation method, something we did not include in previous work on the topic [64].

## A.1 Diagonalising the Hamiltonian

We start from a Hamiltonian describing a one-dimensional chain of non-interacting fermions of length $L$:

$$H(l) = \sum_i h_i(l) n_i + \sum_{i \neq j} J_{ij}(l) c_i^\dagger c_j, \tag{52}$$

where the coefficients $h_i(l=0)$ and $J_{ij}(l=0)$ are arbitrary. We split this into the diagonal component $H_0(l) = \sum_i h_i(l) n_i$ and off-diagonal component $V(l) = H(l) - H_0(l)$. The canonical generator for this problem is given by:

$$\eta(l) = [H_0(l), V(l)] = \sum_{ij} J_{ij}(l)(h_i(l) - h_j(l)) c_i^\dagger c_j. \tag{53}$$

We can compute the flow of the Hamiltonian from $\mathrm{d}H/\mathrm{d}l = [\eta(l), H(l)]$ and read off the flow equations for the running coupling constants:

$$\frac{\mathrm{d}J_{ij}}{\mathrm{d}l} = -J_{ij}(h_i - h_j)^2 - \sum_k J_{ik} J_{kj}(2h_k - h_i - h_j), \tag{54}$$

$$\frac{\mathrm{d}h_i}{\mathrm{d}l} = 2 \sum_j J_{ij}^2 (h_i - h_j), \tag{55}$$

where we have suppressed the dependence on the flow time $l$ for clarity. A detailed introduction to the flow equation method as applied to a system of non-interacting fermions may be found in Ref. [64].

## A.2 Obtaining the eigenstates

The procedure for obtaining the eigenstates is simpler for static systems than for Floquet systems. We apply the same logic, namely the fact that the eigenstates are the single-particle states of the diagonal Hamiltonian, and make an ansatz for the form of the running creation operator:

$$c_i^\dagger(l) = \sum_j \alpha_{i,j} c_k^\dagger, \tag{56}$$

again with $\alpha_{i,j}(l=\infty) = \delta_{i,j}$. From this, we can obtain the flow equation:

$$\frac{\mathrm{d}c_i^\dagger}{\mathrm{d}l} = [\eta(l), c_i^\dagger(l)] = \sum_{jk} J_{jk}(h_j - h_k) \alpha_{i,k} c_j^\dagger, \tag{57}$$

which allows us to compute the single-particle eigenstates in the microscopic $l=0$ basis. This procedure can also be generalised to interacting systems with additional approximations.

# B  Details of the calculation

A few useful identities that we will need are as follows:

$$\left[ c_\alpha^\dagger c_\beta, c_\gamma^\dagger c_\delta \right] = c_\alpha^\dagger \{c_\beta, c_\gamma^\dagger\} c_\delta - c_\gamma^\dagger \{c_\delta, c_\alpha^\dagger\} c_\beta = \delta_{\beta\gamma} c_\alpha^\dagger c_\delta - \delta_{\delta\alpha} c_\gamma^\dagger c_\beta, \tag{58}$$

$$\hat{\sigma}_n \hat{\sigma}_m = \hat{\sigma}_{n+m}, \tag{59}$$

$$[\hat{\sigma}_n, \hat{\sigma}_m] = 0, \tag{60}$$

$$[\hat{n}, \hat{\sigma}_n] = n\hat{\sigma}_n, \tag{61}$$

$$[A \otimes B, C \otimes D] = (AC) \otimes (BD) - (CA) \otimes (DB). \tag{62}$$

## B.1 Defining and Computing the Generator

Following Eq. 19, we can write the Floquet operator explicitly as $K = K_0 + K_{off}$, with:

$$K_0 = \left[ \sum_i h_i^{(0)} c_i^\dagger c_i \right] \otimes \mathbb{1} + \mathbb{1} \otimes \omega \hat{n}, \tag{63}$$

$$K_{off} = \sum_{n \neq 0} \sum_i h_i^{(n)} c_i^\dagger c_i + \sum_n \sum_{ij} J_{ij}^{(n)} c_i^\dagger c_j \otimes \hat{\sigma}_n, \tag{64}$$

where the superscripts are harmonic indices. From this, the generator is defined as $\eta = [K_0, K_{off}]$. The calculation of the generator can be separated into two non-zero parts:

$$i) \quad \left[ \sum_k h_k^{(0)} c_k^\dagger c_k \otimes \mathbb{1}, \sum_n \sum_{ij} J_{ij}^{(n)} c_i^\dagger c_j \otimes \mathbb{1} \right] = \sum_{ij} J_{ij}^{(n)} (h_i^{(0)} - h_j^{(0)}) c_i^\dagger c_j \otimes \hat{\sigma}_n, \tag{65}$$

$$ii) \quad [\mathbb{1} \otimes \omega \hat{n}, K_{off}] = \sum_{n \neq 0} \omega n \left[ \sum_i h_i^{(n)} c_i^\dagger c_i + \sum_{ij} J_{ij}^{(n)} c_i^\dagger c_j \right] \otimes \hat{\sigma}_n, \tag{66}$$

which, when put back together, reduce to $\eta = [K_0, K_{off}] = \eta_1 + \eta_2$ with:

$$\eta_1 = \sum_n \left( \sum_{ij} J_{ij}^{(n)} (h_i^{(0)} - h_j^{(0)}) c_i^\dagger c_j \right) \otimes \hat{\sigma}_n, \tag{67}$$

$$\eta_2 = \sum_{n \neq 0} \omega n \left( \sum_i h_i^{(n)} c_i^\dagger c_i + \sum_{ij} J_{ij}^{(n)} c_i^\dagger c_j \right) \otimes \hat{\sigma}_n, \tag{68}$$

## B.2 Computing the flow of the running couplings

The calculation of the flow of the Floquet operator $dK/dl = [\eta, K]$ can be separated into four pieces:

$$i) \quad [\eta_1, \mathbb{1} \otimes \omega \hat{n}] = \left[ \sum_n \left( \sum_{ij} J_{ij}^{(n)} (h_i^{(0)} - h_j^{(0)}) c_i^\dagger c_j \right) \otimes \hat{\sigma}_n, \mathbb{1} \otimes \omega \hat{n} \right]$$

$$= -\sum_n n\omega \sum_{ij} J_{ij}^{(n)} (h_i^{(0)} - h_j^{(0)}) c_i^\dagger c_j \otimes \hat{\sigma}_n, \tag{69}$$

$$ii) \quad \left[ \eta_1, \sum_m \sum_{ij} H_{ij}^{(m)} \otimes \hat{\sigma}_m \right]$$

$$= \left[ \sum_n \left( \sum_{ij} J_{ij}^{(n)} (h_i^{(0)} - h_j^{(0)}) c_i^\dagger c_j \right) \otimes \hat{\sigma}_n, \sum_m \left( \sum_k h_k^{(m)} c_k^\dagger c_k + \sum_{kq} J_{kq}^{(m)} c_k^\dagger c_q \right) \otimes \hat{\sigma}_m \right]$$

$$= \sum_{n,m} \sum_{ij} J_{ij}^{(n)} (h_i^{(0)} - h_j^{(0)}) (h_j^{(m)} - h_i^{(m)}) c_i^\dagger c_j \otimes \hat{\sigma}_{n+m}$$

$$+ \sum_{n,m} \sum_{ijk} J_{ik}^{(n)} J_{kj}^{(m)} (h_i^{(0)} + h_j^{(0)} - 2h_k^{(0)}) c_i^\dagger c_j \otimes \hat{\sigma}_{n+m}, \tag{70}$$

$$iii) \quad [\eta_2, \mathbb{1} \otimes \omega \hat{n}] = \left[ \sum_{n \neq 0} \omega n \left( \sum_i h_i^{(n)} c_i^\dagger c_i + \sum_{ij} J_{ij}^{(n)} c_i^\dagger c_j \right) \otimes \hat{\sigma}_n, \mathbb{1} \otimes \omega \hat{n} \right]$$

$$= -\sum_{n \neq 0} n^2 \omega^2 \left[ \sum_i h_i^{(n)} c_i^\dagger c_i + \sum_{ij} J_{ij}^{(n)} c_i^\dagger c_j \right] \otimes \hat{\sigma}_n, \tag{71}$$

$$iv) \quad \left[ \eta_2, \sum_m \sum_{ij} H_{ij}^{(m)} \otimes \hat{\sigma}_m \right]$$

$$= \left[ \sum_n \omega n \left( \sum_i h_i^{(n)} c_i^\dagger c_i + \sum_{ij} J_{ij}^{(n)} c_i^\dagger c_j \right) \otimes \hat{\sigma}_n, \sum_m \left( \sum_k h_k^{(m)} c_k^\dagger c_k + \sum_{kq} J_{kq}^{(m)} c_k^\dagger c_q \right) \otimes \hat{\sigma}_m \right]$$

$$= \sum_{n,m} n\omega \sum_{ij} \left[ J_{ij}^{(m)}(h_i^{(n)} - h_j^{(n)}) + J_{ij}^{(n)}(h_j^{(m)} - h_i^{(m)}) + \sum_k (J_{ik}^{(n)} J_{kj}^{(m)} - J_{ik}^{(m)} J_{kj}^{(n)}) \right] c_i^\dagger c_j \otimes \hat{\sigma}_{n+m}. \tag{72}$$

Putting these terms back together, we obtain the flow equations shown in the main text.

### B.3 Interacting Systems

In the case of the interacting system, again following Eq. 19, we can write the Floquet operator as $K = K_0 + K_{\text{off}}$, with two new additional terms due to the interactions:

$$K_0 = \left[ \sum_i h_i^{(0)} c_i^\dagger c_i \right] \otimes \mathbb{1} + \mathbb{1} \otimes \omega \hat{n} + \sum_{ij} \frac{\Delta_{ij}^{(0)}}{2} n_i n_j \otimes \mathbb{1}, \tag{73}$$

$$K_{\text{off}} = \sum_{n \neq 0} \sum_i h_i^{(n)} c_i^\dagger c_i + \sum_n \sum_{ij} J_{ij}^{(n)} c_i^\dagger c_j \otimes \hat{\sigma}_n + \sum_n \sum_{ij} \frac{\Delta_{ij}^{(n)}}{2} n_i n_j \otimes \hat{\sigma}_n. \tag{74}$$

The calculation proceeds similarly to the non-interacting case, but now the generator in turn acquires two additional terms:

$$\eta_3 = \left[ \frac{1}{2} \sum_{ij} \Delta^{(0)} n_i n_j \otimes \mathbb{1}, \sum_n J_{ij}^{(n)} c_i^\dagger c_j \otimes \hat{\sigma}_n \right] = \sum_n \sum_{ijk} J_{ij}^{(n)}(\Delta_{ik}^{(0)} - \Delta_{jk}^{(0)}) c_k^\dagger c_k c_i^\dagger c_j \otimes \hat{\sigma}_n, \tag{75}$$

$$\eta_4 = \left[ \mathbb{1} \otimes \omega \hat{n}, \sum_{n \neq 0} \frac{1}{2} \Delta_{ij}^{(n)} n_i n_j \otimes \hat{\sigma}_n \right] = \sum_{n \neq 0} \frac{\omega n}{2} \sum_{ij} \Delta_{ij}^{(n)} n_i n_j \otimes \hat{\sigma}_n, \tag{76}$$

and all other new terms evaluate to zero. These new generator terms lead to new terms in the flow of the Floquet operator $\mathrm{d}K/\mathrm{d}l = [\eta, K]$ which contribute towards the renormalization of

the interaction coefficient, and are given by:

$$
\begin{aligned}
[\eta_3, K] =& \sum_{nm}\sum_{ijk} J_{ij}^{(n)}(\Delta_{ik}^{(0)} - \Delta_{jk}^{(0)})(h_j^{(m)} - h_i^{(m)})c_k^\dagger c_k(c_i^\dagger c_j + c_j^\dagger c_i) \otimes \hat{\sigma}_{n+m} \\
&+ \frac{1}{2}\sum_{nm}\sum_{ijkl} J_{ij}^{(n)} J_{kl}^{(m)}(\Delta_{ik}^{(0)} - \Delta_{jk}^{(0)})(c_k^\dagger c_l - c_l^\dagger c_k)(c_i^\dagger c_j - c_j^\dagger c_i) \otimes \hat{\sigma}_{n+m} \\
&+ \frac{1}{2}\sum_{nm}\sum_{ijk} J_{il}^{(n)} J_{lj}^{(m)}(\Delta_{ik}^{(0)} + \Delta_{jk}^{(0)} - 2\Delta_{lk}^{(0)})c_k^\dagger c_k(c_i^\dagger c_j + c_j^\dagger c_i) \otimes \hat{\sigma}_{n+m} \\
&- \sum_n \frac{\omega n}{2}\sum_{ijk} J_{ij}^{(n)}(\Delta_{ik}^{(0)} - \Delta_{jk}^{(0)})c_k^\dagger c_k(c_i^\dagger c_j - c_j^\dagger c_i) \otimes \sigma_n,
\end{aligned}
\tag{77}
$$

$$
[\eta_4, K] = \sum_n \frac{\omega^2 n^2}{2}\sum_{ij} \Delta_{ij}^2 n_i n_j \otimes \hat{\sigma}_n + \sum_{nm}\sum_{ijk} J_{ij}^{(m)}(\Delta_{ik}^{(n)} - \Delta_{jk}^{(n)})c_k^\dagger c_k(c_i^\dagger c_j - c_j^\dagger c_i) \otimes \hat{\sigma}_{n+m},
\tag{78}
$$

where we have neglected terms containing more than four fermionic operators. Note that these expressions were obtained using the normal-ordering scheme set out in Refs. [61,65], in order to enforce consistent ordering of fermionic operators throughout the calculation, with the normal-ordering corrections then set to zero at the end. Discarding all other newly generated terms outside of the manifold of Eq. 45, i.e. terms where the operators are not of the form $n_i n_j$, these new contributions lead to the flow equation for the interaction term:

$$
\frac{\mathrm{d}\Delta_{ij}^{(n)}}{\mathrm{d}l} = -\omega^2 n^2 \Delta_{ij}^{(n)} + 2\sum_m \sum_{k\neq i,j}\left[ J_{ik}^{(n-m)} J_{ik}^{(m)}(\Delta_{ij}^{(0)} - \Delta_{kj}^{(0)}) + J_{jk}^{(n-m)} J_{jk}^m(\Delta_{ij}^{(0)} - \Delta_{ik}^{(0)})\right].
\tag{79}
$$

# C  Level-Statistics crossover in the driven Anderson model

The driven Anderson model considered in the main text (Section 5) differs from others found in the literature in the choice of drive. Rather than applying a periodic modulation to any individual term of the Hamiltonian, we instead separate the Hamiltonian into two parts and apply them sequentially over a single cycle of period $T$. We refer to this as a 'step drive' protocol, and it is illustrated in Fig. 2.

In the high-frequency limit, the behaviour of the driven system will be governed by a Floquet Hamiltonian which is given by the time-average of Eq. 27. This effective time-independent Hamiltonian is therefore itself an Anderson Hamiltonian, and consequently the system will exhibit localized behaviour, and the emergence of Poissonian level spacing statistics. In the low-frequency limit, however, things are different: as the hopping term and disordered potential are never applied simultaneously, the instantaneous behaviour is not simply given by a standard Anderson Hamiltonian (in fact, it may be shown numerically that the effective Hamiltonian is not given by the time-average of Eq. 27), and therefore we may expect richer behaviour. Specifically, in the main text we showed that both exact diagonalization (ED) and flow equations (FE) predict a crossover to Circular Ensemble level spacing statistics at low frequency, a property characteristic of delocalization.

To further investigate the fate of this crossover upon increasing the system size in this Appendix we briefly present further results on the Anderson model in the presence of two different drive protocols. In order to demonstrate that the behaviour seen in the main text is not an artefact of the flow equation method, we use exact diagonalization to obtain the single particle level statistics of the Anderson model with a step-like drive (as detailed in the main text) on a variety of system sizes. The results are shown in Fig. 10(a). We see that the

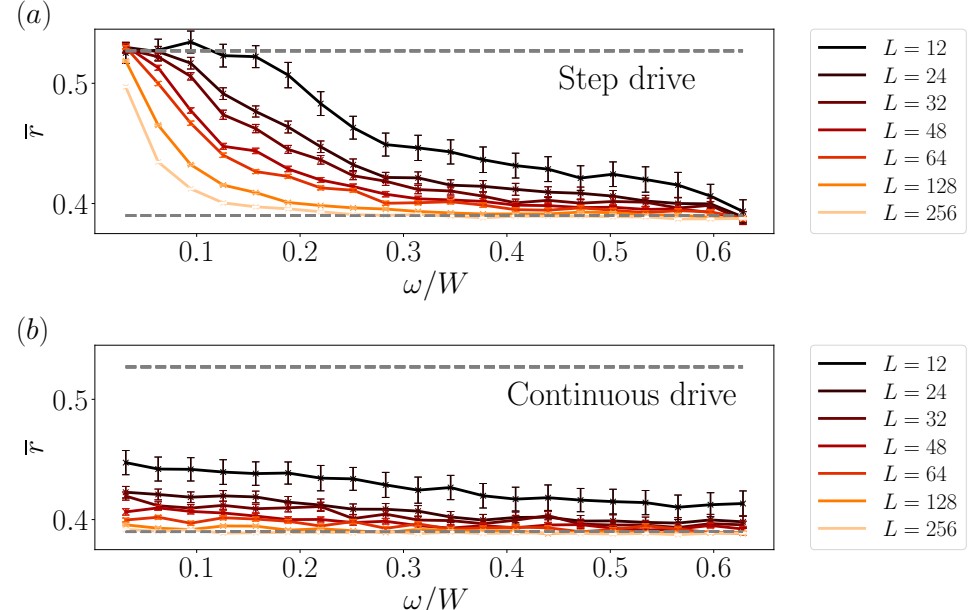

Figure 10: (a) Single-particle level spacing statistics for the driven Anderson model with a step drive protocol, shown for a variety of system sizes $L$ with disorder strenth $W = 5$ and $J_0 = 0.5$, as used throughout the main text. These results were averaged over $N_s = 2048$ disorder realisations for $L = 12$, $N_s = 1024$ disorder realisations for $L = 24$ and $N_s = 512$ disorder realisations for all other system sizes. Error bars show the variance across disorder realisations. There is a clear crossover from Poissonian level statistics ($\overline{r} \approx 0.39$) at high frequency to Circular Ensemble statistics ($\overline{r} \approx 0.53$) at low frequency. The critical frequency for this crossover decreases for larger systems. (b) Single-particle level spacing statistics for the driven Anderson model with a continuous, monotonic drive protocol, with all other parameters the same as in panel (a). Note that in this case, there is no crossover to Circular Ensemble statistics at low frequency, and the system remains localized for all values of $\omega/W$.

frequency below which the finite size level spacing crosses over to the Circular Ensemble value decreases with $L$ and that at the largest size considered ($L = 256$) the level statistics remain close to the localized Poisson limit for a large range of frequencies. This suggests that the observed crossover disappears in the thermodynamic limit and the system remains localized for all frequencies. Such result seems compatible with the available literature on the driven Anderson model [89–91]. We notice, however, that quite interestingly the behavior of the level statistics depends strongly on the nature of the periodic drive. To show this, we further compare the step-drive used in the main text with a monochromatic sinusoidal modulation (where we choose $F(t) = \cos(\omega t)$ and $G(t) = 1 - F(t)$ in Eq. 27), shown in Fig. 10(b). We find that for all system sizes we have investigated, the level statistics remains close to the localized value and no crossover to a delocalized regime emerges.

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
