# Peer review of "Flow Equations for Disordered Floquet Systems"

_SciPost Physics, doi:SciPost Phys. 11, 028 (2021)_

## Round 1 · Referee Report · Anonymous (Referee 1) · 2020-10-15

Strengths
1) Very well written introduction to Wegner's flow equation method applied to (disordered) Floquet systems, paedagogical with all the necessary details, very good literature overview 2) Interesting outlook for future work using this method for other models, especially beyond 1d
Weaknesses
1) Erroneous identification (at least to my understanding) of a delocalization-localization transition in the driven Anderson insulator 2) Missing discussion of resonance problem for driven interacting system
Report
This manuscript gives a very good and clearly written introduction to Wegner’s flow equation method applied to disordered Floquet driven systems. Floquet driven systems are of considerable current experimental and theoretical interest (Floquet topological states of matter, quantum time crystals, etc.). There is a need for new theoretical methods to deal with interactions in such driven systems, especially away from the well studied high frequency limit and/or beyond 1d. The flow equation machinery has the potential to be a helpful new tool for such investigations.
The authors provide a comprehensive literature overview and clearly compare different flow generators, which are at the “heart” of the flow equation method. All the necessary calculations are spelled out, which makes it easy for others to apply their approach to different problems.
The authors then discuss two applications of their approach to specific models, namely the driven Anderson model and weakly interacting disordered fermions. In both applications there are issues that need to be addressed.
1) Driven Anderson model: 1a) The authors discuss the 1d model and state that they find a transition as a function of driving frequency to a delocalized phase. To my understanding this is inconsistent with the literature. The Floquet-Hamiltonian is quasi one-dimensional and even weak disorder will lead to localization. As discussed in Ref. [88] the dynamics appears diffusive over a single drive cycle, but the system remains localized. How does this fit the numerical results in this paper? Also, one really needs to look at different system sizes L before drawing any conclusions. The authors only vary N_h which defines the truncation in Sambe space but not L. Why? Larger systems are discussed for the interacting model, but not here for the non-interacting model. These points need to be clarified/improved/corrected. 1b) Does one have convergence when the driving frequency lies in the disorder bandwidth? For sufficiently large systems one might expect to find resonances in individual disorder realizations, which lead to degeneracies that cannot be resolved by flow equations. The authors should comment on this point since the analysis and proliferation of such resonances is really the key issue in Refs. [87-89].
2) Weakly interacting fermions:
2a) The authors use a weak coupling expansion in (39). However, one needs to be very careful because higher order coupling terms contain (higher order) energy denominators, which can invalidate the expansion even for small coupling in case of resonances from the driving. The authors should comment on this admittedly very difficult point (pointing out the possibility of such problems would already be sufficient, everything else can be considered beyond the scope of this manuscript).
2b) In Fig. 6b it is really not visible that there are significant deviations between ED and flow equations. One needs to present the data differently if one discusses the (not unexpected) deviations for low frequencies.
In summary this is an interesting manuscript. However, the above points need to be resolved before publication, especially 1a).

Author: Steven Thomson on 2020-12-23 [id 1103]
(in reply to Report 1 on 2020-10-15)We thank the Referee for their careful reading of our manuscript, and we respond in detail the points raised below.
1a) "The authors discuss the 1d model and state that they find a transition as a function of driving frequency to a delocalized phase. To my understanding this is inconsistent with the literature. The Floquet-Hamiltonian is quasi one-dimensional and even weak disorder will lead to localization. As discussed in Ref. [88] the dynamics appears diffusive over a single drive cycle, but the system remains localized. How does this fit the numerical results in this paper? Also, one really needs to look at different system sizes L before drawing any conclusions. The authors only vary $N_h$ which defines the truncation in Sambe space but not L. Why? Larger systems are discussed for the interacting model, but not here for the non-interacting model. These points need to be clarified/improved/corrected. "
Response:
We thank the referee for this comment and for drawing our attention to several previous works on driven Anderson Insulators, notably Refs. [88-90] of the revised manuscript.
To answer the Referee's main concern, namely that our results are not consistent with existing literature, we highlight the fact that our driving protocol (a step drive, or bang-bang protocol) differs from the traditional monochromatic drive considered in the references above. This has important consequences on the physics of the problem, especially at intermediate and low frequencies. In fact, within our protocol, the effective Floquet Hamiltonian to which the Referee points is a one-dimensional Anderson model only in the high-frequency limit, where indeed our results show localisation.
To further substantiate this point, we have computed (using exact diagonalisation, to demonstrate that this is not a feature of our flow equation method) the level spacing statistics as a function of drive frequency for both monochromatic drive ($F(t)=\cos(\omega t)$) and a step-like drive ($F(t) = \textrm{sign}[\cos(\omega t)]$), for different system sizes (see new Appendix C). We find that the step-like drive exhibits the delocalisation transition discussed in our manuscript, while monochromatic drive (as considered in previous works) does not, in agreement with the literature.
Finally, we emphasize that the main point behind Section 5 of the manuscript was to benchmark the new flow equation approach against exact diagonalisation (ED), rather than provide a detailed discussion of the driven Anderson model in full generality. For these reasons we consider rather small system sizes ($L=12$) and focus on the level statistics, which is particularly challenging to obtain numerically with flow equations since very long flow times are required. A more detailed discussion of the properties of a driven Anderson model is beyond the scope of this work.
1b) "Does one have convergence when the driving frequency lies in the disorder bandwidth? For sufficiently large systems one might expect to find resonances in individual disorder realizations, which lead to degeneracies that cannot be resolved by flow equations. The authors should comment on this point since the analysis and proliferation of such resonances is really the key issue in Refs. [87-89]."
Response:
We believe that this point is already answered by our Fig. 4, which compares the relative error (panel a) in the FE quasienergies measured with respect to the ED results, and the level spacing statistics computed by both ED and FE methods (panel b) for a variety of values of drive frequency at fixed disorder strength W=5. This figure demonstrates that by retaining a sufficiently high number of harmonics, we obtain good convergence to the numerically exact results at all frequencies we have considered, including frequencies smaller than the disorder bandwidth. We have explicitly rewritten all mentions of the drive frequency $\omega$ in dimensionless form, normalised by the disorder bandwidth, as $\omega/W$ in order to make this clearer throughout.
Flow equations methods in disordered systems typically have no significant problems with resonances in the normal sense of the word, e.g. situations where the on-site energy difference is smaller than the hopping $J_{ij} \geq (h_i - h_j)$. In the limit of $l \to \infty$, only an exact degeneracy between the on-site energies of two sites will lead to the failure of an off-diagonal element to decay quickly, however as such degeneracies are never exact to numerical precision, what we instead observe in practice is the asymptotically slow convergence of the system with increasing flow time. As such, this problem can be mitigated by using a sufficiently large flow time. In practice, we do not find this to be an issue. We note that in previous works (e.g. Ref. [60]), the insensitivity of flow equation methods to resonances has already been remarked upon, even in interacting systems.
2a) "The authors use a weak coupling expansion in (39). However, one needs to be very careful because higher order coupling terms contain (higher order) energy denominators, which can invalidate the expansion even for small coupling in case of resonances from the driving. The authors should comment on this admittedly very difficult point (pointing out the possibility of such problems would already be sufficient, everything else can be considered beyond the scope of this manuscript). "
Response:
We believe the referee may have misunderstood certain aspects of our technique. We do not use a weak coupling expansion in Eq. 39, and there are no ‘energy denominators’ at any point in our calculation. The fixed-point Hamiltonian for weak interactions is not the result of perturbation theory, but rather based around a truncation in operator space. The calculation is based on earlier work in Refs. [61] and [65] where this truncated approach was originally developed for many-body localized systems in the localized regime. We have added a new subsection to Appendix B outlining the calculation of the flow equation for the interacting terms (Appendix B3) in order to clearly demonstrate the technique.
There is no significant issue with resonances in the flow equation method, as discussed in Ref. [60] for the case of a static many-body localized system. To quote Ref. [60], Wegner-type flows "only slow down when the problem is nearly diagonal, blithely integrating past would-be resonances that complicate ordinary perturbative treatments". In truncated flow equation methods of the type used in the present work, the main problem is runaway exponential increase of the interaction terms in the delocalized phase due to the structure of the nested commutators. As we discussed in Section 6 of the original manuscript, this is a problem which can be ameliorated in future work by extending the method proposed here to include more advanced techniques such as normal-ordering corrections which couple the flow of the interaction terms to the flow of the quadratic terms and prevent such divergences from occurring. We believe this discussion to be outwith the scope of the current manuscript, but provide several references in the text to other works discussing this in more detail. Here, we restrict to the weakly-interacting regime where such problems are minimized.
2b) "In Fig. 6b it is really not visible that there are significant deviations between ED and flow equations. One needs to present the data differently if one discusses the (not unexpected) deviations for low frequencies."
Response:
We have added an additional figure (Fig. 7 of the revised manuscript) to address this issue and now present the data for the interacting system in a similar manner as the non-interacting system. We have replaced the left column of Fig. 6 with data at an even lower drive frequency where the disagreement between FE and ED methods is clearly visible.
Additionally, we have replaced Figs. 3, 4, 5, 6 and 8 with new data, incorporating a greater number of disorder averages and a wider range of frequencies, as well as other minor cosmetic improvements throughout the manuscript.

---

## Round 2 · Author Response

We thank the Referee for their careful reading of our manuscript, and are pleased that they find it very clearly written and interesting.

We shall respond to the Referee’s points in detail in our Author Reply, however we wish to clearly state here that our identification of the delocalization transition in the driven Anderson insulator is not erroneous, or in conflict with any of the existing literature: the previous studies that the Referee cites used a monochromatic drive, however in the present work we use a step-like drive which has dramatically different effects.

In our revised manuscript, we clearly address this discrepancy with an additional new Appendix C. We have also added further discussion about resonances in both the interacting and non-interacting models, and have improved the clarity and presentation of our results throughout.

---

## Round 2 · List of Changes

i) We have added a new figure (Fig. 7 of the revised manuscript) which quantitatively demonstrates how the relative error in the many-body quasienergies changes with drive frequency.

ii) Added an Appendix B3 which shows the calculation for the flow of the interaction terms.

iii) New Appendix C, including a new Fig. 9, which discusses the differences between monochromatic and step-like drive protocols, with new numerical results to clearly demonstrate that our findings are consistent with previous work.

iv) New paragraph below Eq. 27 discussing resonances in non-interacting systems, and a brief discussion of resonances in intearacting systems added below Eq. 39.

v) Other minor changes: all figures updated to be in vector graphics format, all frequencies are now normalised by the disorder bandwidth, and other minor typographical edits throughout.

---

## Round 3 · Referee Report · Anonymous · 2021-2-1

Strengths

Very nice and detailed overview of a flow equation method for Floquet systems.

Weaknesses

1. No self consistent way to estimate an error within the method in larger systems (beyond ED) is presented.

2. No examples of non trivial reliable results beyond exact diagonalization

Report

I really enjoyed reading this paper. It is written in a very pedagogical way. I learned a lot. I have several questions/comments. Some of them minor, some are more conceptual. I will list them together in the order they appear in the text.

1) Equation (12). I think the notation is not the best. LHS is a state, RHS is an operator. I do not think this equality is mathematically correct.

2) I am not sure I agree with the discussion after Eqs. (26) and (27). The authors say that $\omega\to 0$ corresponds to the static limit. Is that so? I would say $\omega\to \infty$ does as the authors later explain. I would think the zero frequency limit is ill defined without cutoff in $N_h$, there is an extra infinite sum compared to the static case.

3) I wonder if in these equations there is any advantage in going to the co-moving frame $h_i^{(n)}\to h_i^{(n)} \exp[-n^2 \omega^2 l]$ and similarly for $J_i^{(n)}$. Then there is an explicit exponential decay in all the non-zero terms in the sum and various approximations are more transparent.

4) It is very difficult to interpret different lines in Fig. 3 without legends/explanations. In the printed version in black and white the figure is simply unreadable.

5) It looks to me that the bottom plot in Fig. 4 does not support very well the author's claim. If I only look at the figure then I would guess that the line does not stop at the asymptotes and keep going up (at small $\omega$) and keep decreasing (at large $\omega$). More points are needed to justify the conclusion. Also the choice of colors for the top panel is far from optimal with largest and smallest frequency being identical.

6) There is a mistake in the inline expression for the Wigner-Dyson distribution on page (17). First of all there is a wrong sign and second this is not the Wigner-Dyson distribution, this is its approximation, i.e. it is the Wigner surmise.

7) I am completely lost in the motivation behind Sec. 5.4, i.e. truncating the flow to a single harmonic and the top panel in Fig. 5. To me this approximation strongly devaluates the method. Why not use the same number of harmonics as everywhere. Also as I mention below keeping all the harmonics should allow, I believe, to have an independent way of estimating the error without need of any ED.

8) What is variational in the ansatz (39)? Is it a "better" word for "truncated" or there is some variational optimization involved? What is minimized then? Also later the authors say that this ansatz is valid in the localized phase. Probably they mean "deeply localized" phase.

9) Let me finish with what I think is the biggest issue with the paper. It does not provide any internal way of estimating the error within the method. Sometimes this is impossible and we have to do what the authors do, compare with ED in small systems and hope the error does not change with the system size. But this does not seem to be the case here. E.g. one can properly evolve the eigenstates backwards, as the authors explain, and then compute the variance of the photon number. This should be available for system sizes well beyond ED. One can probably do something different with the same idea. Right now the authors advocate the method as an efficient way to overcome difficulties of ED but never show any advantages as all the examples except the last one are confined to very small systems and the last example does not have any independent error estimate. To me the step of finding a clear and controlled example, where the method goes beyond other methods is a crucial part for advocating that it offers real advantages.

---

## Round 3 · Referee Report · Anonymous · 2021-2-8

Report

This is a well written manuscript that serves as a very good introduction to the flow equation formalism applied to Floquet systems. However, for the same reason it is also necessary to be very careful in describing strengths and weaknesses of the method.

1) My main criticism centers around the sentences:
- Page 13: “There is, however, no divergence encountered due to resonant terms, as in typical perturbative treatments of disordered systems: the sole effect of closely separated on-site energies is the need for a large flow time lmax”
- Page 20: “Note that this truncated form of K(l) is not the result of a perturbative analysis, but instead arises from a truncation in operator space which is valid in the localized phase. Flow equation methods based around Wegner-type generators therefore avoid problems with resonances [60] which can otherwise lead to divergent terms in perturbative treatments of interacting systems” in combination with “variational manifold” below Eq. (39).

Indeed, for a quadratic Hamiltonian like the driven Anderson model (page 13) there are no problems with divergences because the unitary flow is exact in the sense that no higher order terms are being generated. So the only complication due to resonances are long flow times, exactly like the authors say. However, for interacting systems (page 20) the situation is different. The very slow decay of couplings in any given order of the flow equation calculation because of (near) resonances generically leads to large new interactions being generated in the next order. So for the interacting system the situation is worse than just needing long flow times, and the problem with resonances is not avoided (unfortunately). The way to expicitly see this is by going beyond the terms taken into account in the ansatz (39), which is of course a bit of work.

However, on the plus side this does provide a check for the internal consistency of the method if the newly generated and neglected terms indeed remain small. It is not even necessary to take the whole flow of the newly generated terms into account, looking at the source term in their flow equations will be sufficient. One can speculate that normal ordering might be helpful, but there is no obvious reason why this should be the case for a localization-delocalization scenario. For the above reasons the term “variational” is also somehow misleading.

At a minimum the authors should change the sentences on page 13 and 20 to correctly describe the limitation of the method. Additionally, I would encourage them to do an error analysis by looking at the size of the newly generated terms in order to show the internal consistency of the approach. This would also address point 9) raised by the second referee. Once this is done I strongly support publication of this well written manuscript.

2) I appreciate the authors’ clarifying comments regarding the localization-delocalization crossover vs. transition. On page 21 (Sect. 6.3) the authors still use the term transition, this should be changed.

---

## Round 3 · Author Response

We would like to thank the Editor for clarifying a point raised by the Referee that we have misunderstood in our first version of the reply. In our previous response, we focused on demonstrating that the emergence of CE level statistics at low drive frequencies was a genuine feature of the model and a direct consequence of the drive protocol, but we did not address the question of whether this is a true transition in the thermodynamic limit.

We would like to kindly ask the Referee to take into account the following reply in addition to our previous one, specifically concerning the Referee's first comment, which we reproduce below.

1a) "The authors discuss the 1d model and state that they find a transition as a function of driving frequency to a delocalized phase. To my understanding this is inconsistent with the literature. The Floquet-Hamiltonian is quasi one-dimensional and even weak disorder will lead to localization. As discussed in Ref. [88] the dynamics appears diffusive over a single drive cycle, but the system remains localized. How does this fit the numerical results in this paper? Also, one really needs to look at different system sizes L before drawing any conclusions. The authors only vary $N_h$ which defines the truncation in Sambe space but not L. Why? Larger systems are discussed for the interacting model, but not here for the non-interacting model. These points need to be clarified/improved/corrected."

The Referee is correct in noticing that what we had previously identified as a delocalization transition as a function of the drive frequency is in fact a crossover.

In the new Appendix C, following the Referee's suggestion, we present results for the level statistics which show how increasing the system size changes the position of the crossover such that the system in the thermodynamic limit remains localised at all frequencies. We have modified the manuscript accordingly (main text and new Appendix C).

We notice nevertheless that the properties of the finite-size driven Anderson model depend strongly on the nature of the driving protocol. In particular we show in Appendix C that this crossover is absent for a monochromatic drive, where the frequency and size dependence of the level statistics is very weak and the system always remains localised.

---

## Round 3 · List of Changes

Added clarifying remarks about the nature of the delocalization crossover seen in the driven non-interacting system as the system size is changed.

Resubmission 2009.03186v2 on 23 December 2020

---

## Round 4 · Referee Report · Anonymous · 2021-7-22

Report
The authors have taken into account my suggestions/criticisms from the previous reports. I appreciate the new section 6.4 which is very helpful in this respect.
As mentioned in my previous report I am now happy to support publication.

---

## Round 4 · Referee Report · Anonymous · 2021-7-23

Report
The authors addressed all the questions by myself, and I believe by the second referee. I have a few optional further suggestions, which authors might or might not want to take into account. Irrespective of their response I recommend the paper now.
1. The authors make some strong statements about MBL phase, which are not completely important for the rest of the paper. Nevertheless, recently many of the statements were challenged in the literature and many earlier claims were shown to be simply incorrect. E.g. on page 2 of https://arxiv.org/abs/2107.05642 by D. Huse and collaborators they openly challenge validity of a simple l-bit picture of MBL and push the transition boundary to W>7 (instead of previously claimed W=3.5), a soon to be published work by D. Sels extended their method to larger system sizes and pushed the MBL transition point (if any) beyond $W_c>18$, which implies that basically all numerical results reporting MBL were actually finite size effects seen in the ergodic phase. Recently there emerged problems with the RG treatment of the transition as well. I believe the authors might soften a bit the language of their paper separating what was "shown" and what is "believed" to be true by some people.
2. On page (21) when the authors say that their method avoids the problem of small denominators, they might want to comment if it arises e.g. in Fig. 6. If the system sizes are small and $\Delta$ is also small (on the scale of level spacing) this problem might not even show up. I think the authors should be maximally fair about this point.
3. Page 22 "moreso" is written as a single word.
4. When the authors introduce their mistake in Eq. (50), I wonder what the absolute scale represents. I.e. is 0.01 small or large, does it change with the system size etc. To me it is more natural to normalize the mistake by dividing it by $I(\ell=0)$. Then the final value of the mistake tells us by how much off-diagonal elements were reduced by the flow.
5. For static systems an alternative to the flow method based on variational adiabatic transformations was introduced to introduce a non-perturbative SW transformation (https://arxiv.org/pdf/1910.11889.pdf). While there was no explicit discussion of the Floquet extension there it still might be worth commenting about that possibility as it also does not contain small denominators and allows for a systematic expansion.

---

## Round 4 · Author Response

Dear Editor,
Please find attached the revised version of our manuscript "Flow Equations for Disordered Floquet Systems".
We are happy to see that both referees found our manuscript clear and well written, and we appreciate their constructive suggestions. The main feedback on the previous version from both referees was that this work would be significantly stronger with an improved error estimate that did not rely on comparison with exact numerics, such that when we consider larger systems than accessible with exact numerics we can still assess the accuracy of the method in a self-consistent way. We have conducted a detailed and thorough investigation of this point, and the major change in the revised version of our manuscript is the addition of several new sections (Sections 4.3.1 and 6.4) which introduce the concept of flow invariants and demonstrate how they may be used to serve this purpose. We thank the referees for motivating us to address this.
We have additionally made many other smaller changes to the manuscript, detailed in our full response to the Referees below.
Yours sincerely, S. J. Thomson, D. Magano & M. Schiró
--- Response to Second Report of First Referee ---
We thank the Referee for their continued detailed reading of our manuscript and their constructive suggestions.
"1) My main criticism centers around the sentences: - Page 13: “There is, however, no divergence encountered due to resonant terms, as in typical perturbative treatments of disordered systems: the sole effect of closely separated on-site energies is the need for a large flow time lmax” - Page 20: “Note that this truncated form of K(l) is not the result of a perturbative analysis, but instead arises from a truncation in operator space which is valid in the localized phase. Flow equation methods based around Wegner-type generators therefore avoid problems with resonances [60] which can otherwise lead to divergent terms in perturbative treatments of interacting systems” in combination with “variational manifold” below Eq. (39).
"Indeed, for a quadratic Hamiltonian like the driven Anderson model (page 13) there are no problems with divergences because the unitary flow is exact in the sense that no higher order terms are being generated. So the only complication due to resonances are long flow times, exactly like the authors say. However, for interacting systems (page 20) the situation is different. The very slow decay of couplings in any given order of the flow equation calculation because of (near) resonances generically leads to large new interactions being generated in the next order. So for the interacting system the situation is worse than just needing long flow times, and the problem with resonances is not avoided (unfortunately). The way to expicitly see this is by going beyond the terms taken into account in the ansatz (39), which is of course a bit of work.
"However, on the plus side this does provide a check for the internal consistency of the method if the newly generated and neglected terms indeed remain small. It is not even necessary to take the whole flow of the newly generated terms into account, looking at the source term in their flow equations will be sufficient. One can speculate that normal ordering might be helpful, but there is no obvious reason why this should be the case for a localization-delocalization scenario. For the above reasons the term “variational” is also somehow misleading."
The Referee is quite correct to say that divergences may arise due to long flow times, as also seen in Ref. 61 for a time-independent system of disordered, interacting fermions. In our previous response, we focused on issues due to resonances in the sense of energy denominators of the form $1/(E_i-E_j)$ becoming ill-defined when the two energies coincided, and leading to divergent terms. As no such energy denominators occur in this method, this precise problem is not encountered. However, as the Referee notes, at low drive frequencies (or small disorder strengths), the long flow times required can lead to eventual divergences of the interacting terms: the sharp-eyed reader may note that this is visible in Fig. 9 for small distances at the lowest drive frequencies considered, and briefly discussed in the context of normal-ordering corrections. We thank the Referee for pointing out that this is indeed a consequence of (near) resonances, and in our revised manuscript we have given a more precise discussion of this point.
Such problems were encountered in previous work by two of us on localization in time-independent systems (Refs 61 and 65). We found that the generation of new terms tended to remain well-controlled in localized phases, and could be quantified by looking at so-called `invariants of the flow', i.e. quantities which would be conserved by an exact unitary transform, but which are not conserved exactly by our approximate ansatz (Eq. 44). We have now extended the discussion of flow invariants to the Wegner Floquet flow (see the new Section 4.3.1) and have used this as a diagnostic to assess the validity of our ansatz (see also next point). We previously found that normal-ordering is a useful technical tool in taming these divergences as it typically leads to more stable numerical behaviour, as discussed in Refs 61 and 65, though its implementation in the driven case is beyond the scope of the present manuscript.
We have amended the term `variational', as also noted by the second referee.
"At a minimum the authors should change the sentences on page 13 and 20 to correctly describe the limitation of the method. Additionally, I would encourage them to do an error analysis by looking at the size of the newly generated terms in order to show the internal consistency of the approach. This would also address point 9) raised by the second referee. Once this is done I strongly support publication of this well written manuscript."
We have acted on the Referee's suggestion and have included a new Section 6.4 where an error analysis is presented based on the flow invariant, which would be exactly conserved in absence of any truncation of newly generated terms and therefore provides us with an estimate of the internal consistency of the method, along the lines of previous methods used in Refs. 61 and 65. The summary of this analysis is given in the new Figure 8 where the violation from exact invariance, $\delta I$, is plotted for the weakly interacting case as a function of the drive frequency, showing the frequency range in which the method remains controlled. Interestingly this range extends well within the disorder bandwidth and it is not therefore limited to the high-frequency regime.
"2) I appreciate the authors’ clarifying comments regarding the localization-delocalization crossover vs. transition. On page 21 (Sect. 6.3) the authors still use the term transition, this should be changed."
We have removed the word transition' when referring to the delocalized system; we thank the Referee for spotting this. We now use the word
transition' only when discussing the interacting system, where we speculate that the breakdown of our ansatz Hamiltonian may signal the onset of a phase transition, as in the time-independent case (Refs 61 and 65).
--- Response to First Report of Second Referee ---
We thank the Referee for their careful reading of the manuscript, and for their detailed feedback.
"1) Equation (12). I think the notation is not the best. LHS is a state, RHS is an operator. I do not think this equality is mathematically correct."
We thank the Referee for pointing this out, and we have corrected this notation in the revised version.
"2) I am not sure I agree with the discussion after Eqs. (26) and (27). The authors say that $\omega \to 0$ corresponds to the static limit. Is that so? I would say $\omega \to \infty$ does as the authors later explain. I would think the zero frequency limit is ill defined without cutoff in $N_h$, there is an extra infinite sum compared to the static case."
The Referee is correct that this is not, strictly speaking, the static limit and we have amended this sentence accordingly. In the limit of $\omega \to 0$, all terms with an explicit frequency dependence vanish and the flow equations take on the form of the time-independent system, but indeed with an additional sum over the harmonic index n. In this manner, the equations more closely resemble the flow equations for a time-independent two-dimensional system.
"3) I wonder if in these equations there is any advantage in going to the co-moving frame $h(n)i \to h(n)iexp[-n^2 \omega^2l]$ and similarly for $J(n)_i$. Then there is an explicit exponential decay in all the non-zero terms in the sum and various approximations are more transparent."
This is a very interesting suggestion: we thank the Referee for drawing our attention to this possibility, and plan to return to it in a future work.
"4) It is very difficult to interpret different lines in Fig. 3 without legends/explanations. In the printed version in black and white the figure is simply unreadable."
Fig. 3 is intended to give the reader an overview of the behaviour of different classes of terms under the action of the continuous unitary transform, e.g. panels (c) and (c) show the flow of the zero-frequency on-site terms (solid) and hopping terms (dashed), while panels (e) and (f) show the flow of the higher frequency terms. The aim of these figures is to demonstrate that the flow is smooth and well-controlled, and also that the higher harmonics decay quickly to zero at high drive frequencies and more slowly in the case of low drive frequencies.
We do, however, agree with the Referee that Fig. 3 (and the similar Fig. 6) is difficult to read, and so we have streamlined this figure in the revised version, now showing the flow of only a small representative subset of the running couplings. We believe that the revised figure is clearer, while still conveying the same qualitative insight into the behaviour of the flow in different regimes.
"5) It looks to me that the bottom plot in Fig. 4 does not support very well the author's claim. If I only look at the figure then I would guess that the line does not stop at the asymptotes and keep going up (at small $\omega$) and keep decreasing (at large $\omega$). More points are needed to justify the conclusion. Also the choice of colors for the top panel is far from optimal with largest and smallest frequency being identical."
In fact, the central claim of Fig. 4b is that the flow equations and exact diagonalization results match extremely well, as this is a highly non-trivial quantity to compute using continuous unitary transforms. Additional results are shown in Appendix C from exact diagonalization demonstrate the behaviour over a greater range of frequencies and system sizes to illustrate the behaviour across a wider parameter. We believe that the data shown in Fig. 4b is sufficient to demonstrate our main point that flow equations give the same results as numerically exact methods in this regime.
We have nonetheless added an additional data point at lower frequencies to show that the level statistics reach a plateau. For this data point, the flow equations do not reproduce the exact diagonalization result, as a larger number of harmonics would be required to accurately capture the behaviour in this low frequency region. We have also updated the colour scheme of Fig. 4a.
"6) There is a mistake in the inline expression for the Wigner-Dyson distribution on page (17). First of all there is a wrong sign and second this is not the Wigner-Dyson distribution, this is its approximation, i.e. it is the Wigner surmise."
We thank the Referee for pointing this out, and have rectified this error in the revised manuscript.
"7) I am completely lost in the motivation behind Sec. 5.4, i.e. truncating the flow to a single harmonic and the top panel in Fig. 5. To me this approximation strongly devaluates the method. Why not use the same number of harmonics as everywhere. Also as I mention below keeping all the harmonics should allow, I believe, to have an independent way of estimating the error without need of any ED."
The Floquet eigenstates are time-independent, and therefore should not have contributions from higher harmonics. In this section, we are approximating the eigenstates of the block-diagonal time-independent matrices sketched in Fig. 1c, not the full time-dependent eigenstates of Fig. 1a. Strictly speaking, to do so exactly would require two unitary transforms: one `reverse' transform to give us the full time-dependent eigenstates of the Floquet operator shown in Fig. 1a, followed by a second transform into the basis of Fig. 1c to give us the time-independent Floquet eigenstates of the effective Floquet Hamiltonian. We included this section to show that at high frequencies, we can obtain the Floquet eigenstates to high accuracy without requiring this two-step procedure.
"8) What is variational in the ansatz (39)? Is it a "better" word for "truncated" or there is some variational optimization involved? What is minimized then? Also later the authors say that this ansatz is valid in the localized phase. Probably they mean "deeply localized" phase."
This point was also mentioned in the second report of the first referee, and we have removed the word `variational'. This ansatz has been shown to be valid in the localized phase of the corresponding time-independent model, only breaking down close to the transition (see Ref. 61), however has also been used to study the delocalization transition in a time-independent model with long-range couplings (Ref. 65), where it was able to capture the delocalized phase in the weakly-interacting regime.
"9) Let me finish with what I think is the biggest issue with the paper. It does not provide any internal way of estimating the error within the method. Sometimes this is impossible and we have to do what the authors do, compare with ED in small systems and hope the error does not change with the system size. But this does not seem to be the case here. E.g. one can properly evolve the eigenstates backwards, as the authors explain, and then compute the variance of the photon number. This should be available for system sizes well beyond ED. One can probably do something different with the same idea. Right now the authors advocate the method as an efficient way to overcome difficulties of ED but never show any advantages as all the examples except the last one are confined to very small systems and the last example does not have any independent error estimate. To me the step of finding a clear and controlled example, where the method goes beyond other methods is a crucial part for advocating that it offers real advantages."
We thank the Referee for motivating us to further develop our error analysis. To this extent we have extended the concept of flow invariants, conserved quantities of the exact unitary flow used in static (time-independent) applications of the flow equation method, to the Wegner Floquet generator used in this work (see new Sect.4.3.1). This allows us to assess the stability of our method in a case where truncation of the newly generated terms in the flow is necessary, such as in presence of interactions, which would result in a violation of the exact conservation law. In particular, as we now discuss in the new Sec.6.4, by monitoring the flow invariant as a function of the external parameters we can assess the regimes in which our truncated ansatz remains under control, as shown in Figure 8.

---

## Round 4 · List of Changes

New Section 4.3.1 on flow invariants and their uses.
Added brief discussion of the conservation of the flow invariant in Section 5.3 for the non-interacting system.
New Section 6.4 where we show the flow invariant for the interacting system, and added new Fig. 8.
Replaced Figs. 3 and 6 with cleaner figures showing fewer lines.
Corrected the notation in Eq. 12.
Minor changes to notation in Section 4.3 to be consistent with the new Sec. 4.3.1.
Replaced Fig. 4a with new colour scheme, and Fig. 4b now includes an extra point at low frequency.
Added sentence regarding the generation of higher-order terms by the flow, referencing work where this is discussed.
Added an additional citation in the introduction (Ref 73).

---

## Editorial Decision

published